# Effective Latent Differential Equation Models via Attention and Multiple Shooting

**Germán Abrevaya**[1]                                                            *gabrevaya@df.uba.ar*

**Mahta Ramezanian-Panahi**[2, 3]                                      *mahta.ramezanian@mila.quebec*

**Jean-Christophe Gagnon-Audet**[2, 3]              *jean-christophe.gagnon-audet@mila.quebec*

**Pablo Polosecki**[4]                                                           *pipolose@us.ibm.com*

**Irina Rish**[2, 3]                                                                 *irina.rish@mila.quebec*

**Silvina Ponce Dawson**[1, 5]                                           *silvina@df.uba.ar*

**Guillermo Cecchi**[4]                                                       *gcecchi@us.ibm.com*

**Guillaume Dumas**[2, 6, 7]                                             *guillaume.dumas@ppsp.team*

[1] *Universidad de Buenos Aires, FCEyN, Departamento de Física. Buenos Aires, Argentina.*
[2] *Mila - Quebec AI Institute. Montréal, Québec, Canada.*
[3] *Université de Montréal. Montréal, Québec, Canada.*
[4] *IBM Research, T.J. Watson Research Center. Yorktown Heights, New York, USA.*
[5] *CONICET - Universidad de Buenos Aires, Instituto de Física de Buenos Aires (IFIBA). Buenos Aires, Argentina.*
[6] *CHU Sainte-Justine Research Center. Montréal, Québec, Canada.*
[7] *Department of Psychiatry and Addictology, Université de Montréal. Montréal, Québec, Canada.*

**Reviewed on OpenReview:** *https://openreview.net/forum?id=uxNfN2PU1W*

## Abstract

Scientific Machine Learning (SciML) is a burgeoning field that synergistically combines domain-aware and interpretable models with agnostic machine learning techniques. In this work, we introduce GOKU-UI, an evolution of the SciML generative model GOKU-nets. GOKU-UI not only broadens the original model's spectrum to incorporate other classes of differential equations, such as Stochastic Differential Equations (SDEs), but also integrates attention mechanisms and a novel multiple shooting training strategy in the latent space. These modifications have led to a significant increase in its performance in both reconstruction and forecast tasks, as demonstrated by our evaluation on simulated and empirical data. Specifically, GOKU-UI outperformed all baseline models on synthetic datasets even with a training set 16-fold smaller, underscoring its remarkable data efficiency. Furthermore, when applied to empirical human brain data, while incorporating stochastic Stuart-Landau oscillators into its dynamical core, our proposed enhancements markedly increased the model's effectiveness in capturing complex brain dynamics. GOKU-UI demonstrated a reconstruction error five times lower than other baselines, and the multiple shooting method reduced the GOKU-nets prediction error for future brain activity up to 15 seconds ahead. By training GOKU-UI on resting state fMRI data, we encoded whole-brain dynamics into a latent representation, learning a low-dimensional dynamical system model that could offer insights into brain functionality and open avenues for practical applications such as the classification of mental states or psychiatric conditions. Ultimately, our research provides further impetus for the field of Scientific Machine Learning, showcasing the potential for advancements when established scientific insights are interwoven with modern machine learning.

# 1 Introduction

## 1.1 Scientific Machine Learning

Scientific Machine Learning (SciML) is an emerging field that, drawing insights from scientific data, seeks to advance data-driven discovery with an approach that can produce interpretable results (Baker et al., 2019). Its synergistic blend of machine learning and scientific computing based on mechanistic models makes it very powerful for addressing complex problems across all STEM areas and beyond (Willard et al., 2022). Using informed priors, its application is already making key contributions in scientific inference, data analysis, and machine learning enhanced modeling (von Rueden et al., 2023). Recent developments in SciML include various approaches to derive dynamical system models from observational data. The sparse identification of nonlinear dynamics (SINDy) algorithm is one such approach which, leveraging recent advances in sparsity techniques, exploits the observation that only a few important terms dominate the dynamics in most physical systems (Brunton et al., 2016). A different well-established candidate for handling model-agnostic systems is deep reinforcement learning, which has been particularly widely adopted by the fluid mechanics community, with applications in flow control and shape optimization (Martín-Guerrero & Lamata, 2021; Viquerat et al., 2022). Physics-informed neural networks (PINNs) are another approach in which neural networks are trained to solve supervised learning tasks with respect to the laws of physics and can be used to derive data-driven partial differential equations or their solutions (Raissi et al., 2019). The application of PINNs to numerically stiff systems (Wang et al., 2021) is challenging. Universal differential equations (UDEs) are a recent method that not only overcomes the stiffness limitation, but also represents a perfect example of the essence of SciML: use all the prior knowledge and scientific insights available for your problem and fill the missing gaps with machine learning (Rackauckas et al., 2020). The simple yet powerful idea behind UDEs involves using traditional differential equation models with some unknown terms that are substituted by universal function approximators, such as neural networks. These approximators will be learned simultaneously with the equations parameters by using sensitivity algorithms (Ma et al., 2021a) within the differentiable modelling framework (Shen et al., 2023).

The evolution equations that might be derived from the data may not actually correspond to mechanistic models based on first principles. That is, on many occasions, the dynamics take place on a manifold of a lower dimension than the full phase space of the system (Foias et al., 1988). Having reduced equations that describe the evolution on these manifolds is very useful, especially in high-dimensional systems (Lucia et al., 2004). Reduced-order models (ROMs) can be derived from data (Guo & Hesthaven, 2019). In particular, generative adversarial network (GAN) (Goodfellow et al., 2014) approaches have been used to enhance the application of ROMs to simulations of fluid dynamics (Kim et al., 2019; 2022).

## 1.2 Neural Differential Equations

Despite the basic ideas behind differential equations parameterized by neural networks and its connection with deep learning had older roots in literature (Rico-Martinez et al., 1992; 1994; Rico-Martinez & Kevrekidis, 1993; Chang et al., 2018; Weinan, 2017; Haber & Ruthotto, 2017; Lu et al., 2018), the publication of Chen et al. (2018) was a turning point in the young history of SciML. Since then, the topic of *neural differential equations* (neural DEs) has become a field, as stated and evidenced in the comprehensive survey by Kidger (2022). In Chen et al. (2018), by interpreting ResNets (He et al., 2016) as a discrete integration of a vector field with the Euler method, the authors proposed an infinitesimally layered neural network as its continuous limit and modeled it with an ordinary differential equation (ODE) parameterized by a neural network, giving rise to the Neural Ordinary Differential Equation (NODE) models. They also demonstrated that NODEs can be trained by backpropagating through black-box ODE solvers using the adjoint method (Pearlmutter, 1995), making it a memory-efficient model.

Furthermore, Chen et al. (2018) introduced the Latent Ordinary Differential Equations (Latent ODEs), a continuous-time generative model that encodes time series data into a latent space that could potentially capture its underlying dynamics, which are modeled using a NODE. First, the observed time series are encoded in the latent space using a recognition model, typically a Recurrent Neural Network (RNN). The temporal dynamics in the latent space is then modeled using NODEs, and lastly, its solution is decoded

back into the observation space to generate predictions or perform other tasks such as anomaly detection or imputation of missing values.

By using this approach, Latent ODEs can capture the intricate and potentially nonlinear dynamical systems that underlie many real-world time series data. The continuous nature of NODEs allows the model to handle irregularly sampled data (Rubanova et al., 2019), a common challenge in many applications. Additionally, the use of the latent space enables the model to capture complex patterns in high-dimensional data, while still providing a compact and interpretable representation of the underlying dynamics.

### 1.3  GOKU-nets

The work of Linial et al. (2021) builds on the basis of the Latent ODEs model demonstrating that the incorporation of prior knowledge of the dynamics involved in the form of a backbone differential equation structure can increase the performance of a purely agnostic model. They propose another continuous-time generative model called GOKU-nets (which stands for Generative ODE Modeling with Known Unknowns), which are the focus of this paper. This model incorporates a variational autoencoder (VAE) structure with a differential equation to model the dynamics in the latent space. However, in this case, a specific form for the ODE is provided while allowing its parameters (the *known unknowns*) to be inferred. The model is trained end-to-end jointly learning the transformation to the latent space, inferring the initial conditions and parameters of the ODE, which will be integrated afterward; and finally, a last transformation is performed to go back to the input space. In the next section, the details of its architecture will be described in detail. Note that the ODE can be integrated further than the input time span in order to generate an extrapolation, thus becoming a forecast of the future evolution of the time series. The study in Linial et al. (2021) compares GOKU-net with baselines such as LSTM and Latent-ODE in three domains: a video of a pendulum, a video of a double pendulum, and a dynamic model of the cardiovascular system. The authors show that their model significantly outperforms the others in reconstruction and extrapolation capabilities, reduces the size of the required training sets for effective learning, and furthermore, has greater interpretability, allowing for the identification of unobserved but clinically meaningful parameters.

The original GOKU-net model was limited to handling only ODEs. In this work, we expand its capabilities by implementing the model in the Julia Programming Language (Bezanson et al., 2017), leveraging its potent SciML Ecosystem (Rackauckas & Nie, 2017) which enables us to utilize a wide spectrum of differential equation classes (including SDEs, DDEs, DAEs) and a diverse suite of advanced solvers and sensitivity algorithms (Rackauckas et al., 2019; Ma et al., 2021a).

As reported in the literature, the identification of nonlinear dynamical systems, and in particular the gradient-descend based training of neural DE models, can be often challenging due to their highly complex loss landscapes, leading to poor local minima training stagnation (Ribeiro et al., 2020; Turan & Jäschke, 2021). We propose an enhancement to the original GOKU-net architecture which adds attention mechanisms to the main part of the model that infers the parameters of the differential equations. Moreover, to overcome the inherent difficulties of training, we developed a novel strategy to train the GOKU-net based on the multiple shooting technique (Bock & Plitt, 1984; Ribeiro et al., 2020; Turan & Jäschke, 2021) in the latent space. We have evaluated our enhanced model and training strategy on simulated data from a network of stochastic oscillators, specifically Stuart-Landau oscillators, as well as empirical brain data derived from resting state human functional Magnetic Resonance Imaging (fMRI). In both cases, the GOKU-net that fuses multiple shooting and attention, labeled *GOKU-nets with Ubiquitous Inference* (GOKU-UI), outperformed both the base GOKU-net model and the baseline models in terms of reconstruction accuracy, forecasting capability, and data efficiency. The application of GOKU-UI in fMRI data analysis not only demonstrates its superiority in handling complex neural datasets but also highlights its potential in enhancing real-time neuromodulation strategies (Parastarfeizabadi & Kouzani, 2017). We believe that GOKU-UI represents a promising step forward in Scientific Machine Learning, underscoring the rich possibilities that emerge when melding traditional scientific insights with contemporary machine learning techniques.

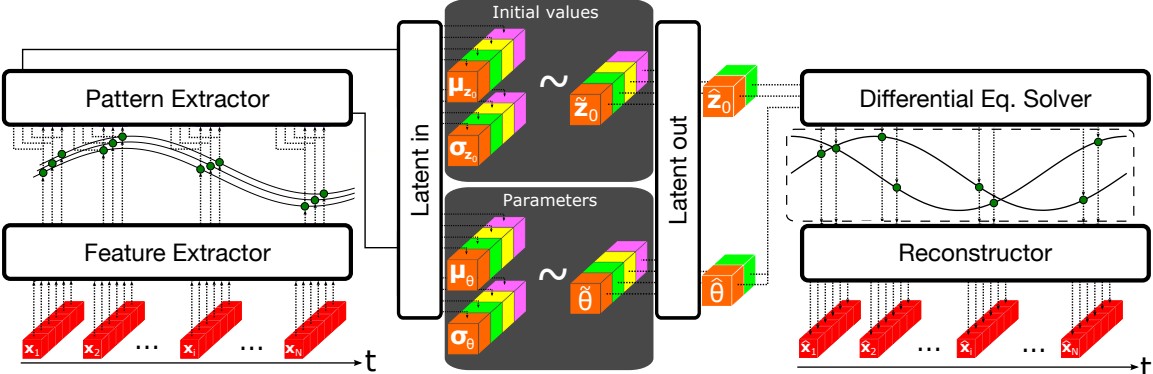

Figure 1: Schematic representation of a general Latent Differential Equation model. First, each time frame of the input data $x_i$ is independently processed by a *Feature Extractor*, then the whole sequence goes through a *Pattern Extractor*, which learns the mean and variance of the initial conditions, and possibly also the parameters, of the differential equation that will be next integrated. Finally, the solution goes through a last transformation performed by a *Reconstructor* in order to go back to the input space.

## 2 Methods

### 2.1 Basic GOKU-nets

GOKU-nets can be considered a specific instance within the broader category of models we denominate Latent Differential Equation models (Latent DEs). These generative models have a structure similar to Variational Autoencoders (VAEs), but with a key distinction: they encode time series data into a latent space governed by differential equations. The objective, akin to any autoencoder model, is to ensure that the output closely reconstructs the input by passing through this latent space. In Latent DEs, the input typically comprises complex, high-dimensional time series data, such as sequential brain images, financial records, or biosignals. After encoding the data into its latent space, beyond simply generating an output that corresponds to the time span of the input, these models can integrate the differential equation further in time, allowing them to forecast the system's future evolution. This capability is not only crucial for validating a model, as is standard in scientific modeling, but also has diverse practical applications. These applications range from finance and weather forecasting to neuromodulation in the context of brain data, where accurate short-term forecasts can help compensate for delays in closed-loop feedback systems (Parastarfeizabadi & Kouzani, 2017).

The schema in Figure 1 illustrates a general Latent DE model. Initially, each temporal frame of the input data $x_i$ is independently processed by a *Feature Extractor*, usually reducing its dimensionality. Following this, the entire sequence is subjected to a *Pattern Extractor*, which aims to learn the distribution of the initial conditions and possibly of the parameters for the differential equation that will be subsequently integrated. Lastly, the solution undergoes a final transformation via a *Reconstructor*, going back to the original input space. The original model is trained as a standard VAE, maximizing the evidence lower bound (ELBO) (Kingma & Welling, 2013).

In the case of the Latent ODEs proposed by Chen et al. (2018), an RNN is used for the Pattern Extractor, a fully connected NN for the Reconstructor, and the differential equation is parametrized with another NN. This means that in this model the specific form of the differential equation is not provided beforehand but is learned simultaneously during training. The Feature Extractor and the intermediate layers, *Latent in* and *Latent out*, are not present so they could be considered as identity operations. On the other hand, the GOKU-net model proposed by Linial et al. (2021) has a ResNet with fully connected NNs as the Feature Extractor, while in the Pattern Extractor, an RNN is used to learn the initial conditions and a bidirectional LSTM for the ODE parameters. In this case the differential equation is explicitly predefined, allowing to incorporate some prior knowledge about the dynamical nature of the system under consideration. The Latent

in and out layers are fully connected NN and finally, the Reconstructor is another ResNet, similar to the initial one. In the next section, some enhancements to the original GOKU-net model are proposed.

## 2.2 GOKU-UI

### 2.2.1 Attention mechanism

The first modification is the addition of a basic attention mechanism (Vaswani et al., 2017) to the Pattern Extractor, specifically in the part associated with the learning of the parameters of the differential equation. Namely, instead of keeping the last element of the bidirectional LSTM (BiLSTM) used in the original GOKU-net model, all of their sequential outputs pass through a dense layer with softmax activation to calculate the attentional scores that would weight the sum of all the BiLSTM outputs in order to obtain its final output.

### 2.2.2 Multiple Shooting

When training Neural DE models, gradients have to be calculated through differential equations with respect to its initial conditions and parameters, by means of some sensitivity algorithm (Ma et al., 2021a). This tends to produce highly complex loss landscapes (Ribeiro et al., 2020; Metz et al., 2021). The work of Turan & Jäschke (2021) demonstrates that training Neural ODEs even on very simple oscillatory data could be problematic, showing that the outcome may result in a trajectory similar to a moving average of the original data, thus failing to capture responses of higher frequency. They proposed a solution based on the *multiple shooting* methods, which are widely used in Optimal Control (Bock & Plitt, 1984; Diehl et al., 2006) and Systems Identification (Baake et al., 1992; Ribeiro et al., 2020) to alleviate the problem of high sensitivity to initial conditions and lower the probability of getting trapped at local minima with very poor performance. The basic idea of multiple shooting is to partition the complete time span over which the differential equation would be integrated into smaller time windows, for each of which the initial conditions are inferred in parallel. Afterwards, the multiple segments are joined and a continuity constraint is imposed during optimization, in our case, through the penalty method, which in practice simply consists of adding a regularization term to the loss function.

However, applying the multiple shooting method to GOKU-nets is not straightforward. Firstly, in most cases that use this method, such as in Turan & Jäschke (2021), the differential equations are typically directly modeling the observable data, having direct access to the true initial conditions for each window. In the case of GOKU-nets, the dynamics modeled by differential equations occur in the latent space, which is being learned simultaneously; as a result, such true initial conditions are not available. Secondly, it is necessary to determine how the method will behave in relation to the parameters of the differential equation, which in the case of Neural ODEs are implicitly learned as part of their parameterization through the neural network.

Our proposal for extending the multiple shooting method to GOKU-nets is as follows. After passing through the Feature Extractor, we divide the temporal interval in the latent space in such a way that the Pattern Extractor generates in parallel different initial conditions for each temporal window, but provides a single set of parameters for the differential equations that will be shared by all windows. By this strategy, we maintain the potential benefits inherent to the multiple shooting method while leveraging the information available in a wider temporal range for the task of parameter inference, which is generally more challenging than estimating initial conditions. As mentioned before, we do not have access to the true initial conditions for grounding the latent trajectories. However, we can strive to achieve continuity across different windows, which is crucial in multiple shooting methods to obtain solutions equivalent to those achieved with single shooting. To this end, these intervals are defined by overlapping the last temporal point of each window with the first one of the following and the goal is to minimize the distance between these points. Specifically, we employ regularization in the cost function when training the model, quadratically penalizing the discrepancy in the latent space of the overlapping points, that is, between the initial condition of each window and the end point of its preceding segment (Vantilborgh et al., 2022; Turan & Jäschke, 2021).

Our experiments indicated that non-variational GOKU-nets models outperform their variational counterparts significantly (see Supplementary Information C). Therefore, we used non-variational GOKU-nets for all the remaining results in this work. Specifically, instead of sampling from normal distributions in the latent space

as shown in Figure 1, we directly take the mean values $\mu_{z_0}$ and $\mu_\theta$. As a result, the cost function associated with these models does not include the KL divergence term associated with the ELBO, but it does retain the reconstruction term, which is calculated as the mean squared error between the model's output and the input, normalized by the mean absolute value of the input. Furthermore, when using multiple shooting training, the continuity regularization described in the previous paragraph is included.

## 2.3 Experiments

In the next sections, we evaluate our proposed attention and multiple shooting enhancements through two highly challenging cases; one on synthetic data based on a network of stochastic oscillators known as Stuart-Landau oscillators and the other on empirical human brain data.

We compare the reconstruction and forecast performance of different variations of the GOKU-model (basic or with attention) trained in the original single shooting fashion or with the proposed multiple shooting method, as well as some baseline models: LSTM, Latent ODE (Chen et al., 2018), Augmented Neural ODE (ANODE) (Dupont et al., 2019) and a naïve model. In the forecast task evaluation, a vector autoregressive (VAR) model (Lütkepohl, 2005) was also included as a baseline, as it is a standard statistical model used for forecasting multivariate time series. For a fair comparison, both the LSTM and Latent ODE models are constructed maintaining the same GOKU-net general architecture and changing only the differential equation layer. Specifically, the Feature Extractor, Pattern Extractor, Latent In, Latent Out, and Reconstructor layers (see Figure 1) maintain the same architecture and hyperparameters. However, the differential equation layer is substituted with a Neural ODE for the Latent ODE model, and in the other case, it is replaced by a recursively executed LSTM. In contrast, the ANODE and VAR directly model the input variables without inferring any latent space representation. The size of the NN parameterizing the differential equation inside the NODE and Latent ODE, as well as its latent state dimensionality and size of the LSTM were selected to match the total number of parameters of their contending GOKU-UI (with attention and multiple shooting). The lag order used in VAR was selected in each case as the one that minimizes the Akaike's Information Criterion) (AIC) on the training data. The naïve predictors, both for the reconstruction and forecast task, are historical averages of the input data, i.e., constant predictions with the mean values of the inputs across time.

All models were trained under identical conditions and following the same procedure, with the aim of minimizing reconstruction error. In all instances, the input sequences to the model consisted of 46 time steps. During each training epoch, a random interval of this length was selected within the training data available for each sample in every batch of 64 samples. In the case of the multi-shooting training, the 46-time step sequence was further partitioned in the latent space into five windows, each comprising 10 time steps, with the last point of one window overlapping with the first point of the subsequent window. Under these circumstances, the loss function was augmented by the sum of squared differences between the endpoints of each segment. This sum was normalized by the number of junctions and multiplied by a regularization coefficient to impose a continuity constraint among the different segments. In the results presented here, the regularization coefficient was set to 2. Comprehensive details of the training process, as well as the specific architecture of the models and the hyperparameters used, can be found in the Supplementary Information.

### 2.3.1 Simulated data

Stuart-Landau (SL) oscillators, representing the normal form of a supercritical Hopf bifurcation, serve as a fundamental archetype of mathematical models and are extensively used across diverse scientific disciplines to study self-sustained oscillations (Kuramoto, 1984; Kuznetsov et al., 1998). The SL oscillator is often described by the complex-valued differential equation

$$\dot{z} = z(a + i\omega) - z|z|^2 \tag{1}$$

where $z = \rho e^{i\theta} = x + iy$, $a$ is the bifurcation parameter, and $\omega$ is intrinsic frequency of the oscillator. The parameter $a$ represents the linear growth rate of the system. When $a$ is positive, the amplitude of the oscillation increases, and when a is negative, the amplitude of the oscillation decreases. At $a = 0$, a Hopf bifurcation occurs, and the system transitions from a stable fixed point to limit cycle oscillations (or vice

versa). Despite its apparent simplicity, the SL model can exhibit a wide range of behaviors, including limit cycles, chaotic oscillations, and quasiperiodicity, making it a versatile tool in the study of nonlinear dynamics and a good candidate for evaluating the capabilities of the GOKU-net models.

In particular, we generate the simulated data with a network of coupled stochastic Stuart-Landau oscillators that has been widely used to model brain dynamics for resting state fMRI (Jobst et al., 2017; Deco et al., 2017; Donnelly-Kehoe et al., 2019; Ipiña et al., 2020), which will also be used in the empirical evaluation on brain data, described in the next section. The dynamics of the $i$-th node within a network of N oscillators is given by the following equation:

$$\dot{x}_j = Re(\dot{z}_j) = [a_j - x_j^2 - y_j^2]x_j - \omega_j y_j + G \sum_{i=1}^{N} C_{ij}(x_i - x_j) + \beta \eta_j(t)$$

$$\dot{y}_j = Im(\dot{z}_j) = [a_j - x_j^2 - y_j^2]y_j - \omega_j x_j + G \sum_{i=1}^{N} C_{ij}(y_i - y_j) + \beta \eta_j(t) \qquad (2)$$

where $C_{ij}$ is a connectivity matrix between all nodes in the network, $G$ is a global coupling factor, while $\eta_j$ represents additive Gaussian noise. Note that independent bifurcation parameters $a_j$ and frequencies $\omega_j$ are used for each node.

During the construction of our dataset, we perform a dimensionality augmentation on the network of oscillators, which are utilized as latent dynamics. Specifically, we apply a fixed random linear transformation, $f : \mathbb{R}^{2N} \to \mathbb{R}^D$, to the latent trajectories of each sample, where the dimension $D$ is much larger than $2N$. All the synthetic data experiments were performed using $N = 3$ stochastic oscillators with a high dimension $D = 784$. Experiments with various sizes of training sets were performed, with training sets ranging from 75 to 4800 different samples. Each sample corresponds to a different set of initial conditions and parameters for the SL differential equations. A validation set of another 200 samples was used for the training termination criteria, and a separate testing set containing 900 different samples was employed for the evaluation. All details of the implementation and hyperparameters can be found in the Supplementary Information.

### 2.3.2 Empirical evaluation

Next, in an effort to evaluate both our proposed models in a challenging empirical dataset and to provide an example of the advantages of incorporating prior scientific insights into agnostic AI models, we focus on one of the most complex systems in nature: the human brain.

We used the resting state fMRI data from 153 subjects, sourced from the Track-On HD study (Klöppel et al., 2015). The data was pre-processed as described in Polosecki et al. (2020), followed by a 20-component Canonical ICA (Varoquaux et al., 2010). Of the original 20 components, 9 were identified as artifacts and therefore eliminated, leaving 11 components for further analysis. The contribution of each subject was captured in two separate visits, yielding an aggregate of 306 data samples. These samples, with 160 time points each, were acquired at a temporal resolution of 3 seconds. In contrast to the simulated data scenario, the empirical data was considerably limited. Consequently, we optimized our data splitting approach. To avoid 'double-dipping', we ensured that the training and testing datasets comprised entirely distinct samples. However, for the training and validation sets, we utilized the same samples, implementing a temporal split. Specifically, approximately 20% of the data samples (n=60) were reserved for testing, while ensuring a balanced representation of the sex, condition, and measurement site. The rest of the data samples (n=246) were allocated for training and validation purposes. Within this subset, the initial 114 time points were designated for model training. The remaining data served for validation and for an criterion early training termination. For a visual representation of the dataset partitioning, please refer to Figure 5 in the Supplementary Information.

Our proposed GOKU-UI model was trained using 20 Stuart-Landau stochastic oscillators (Eq. 2) in the latent space. A comparison of the model's performance using various numbers of oscillators and a thorough description of the training procedure are presented in the Supplementary Information section.

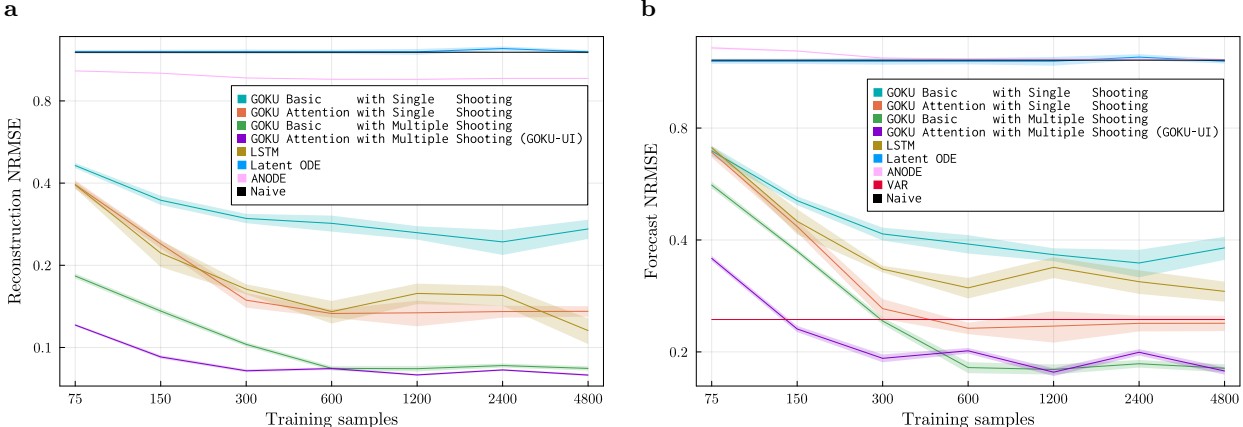

Figure 2: Comparison of the reconstruction (a) and forecast (b) performances on the synthetic Stuart-Landau dataset measured by the median normalized RMSE while increasing the number of samples in the training set. The averages were taken with respect to the input dimensions and time span, while the shaded regions correspond to standard errors derived from various training runs with multiple random seeds. The forecast was evaluated on a 20 time-steps horizon.

## 3 Results

We evaluated the performance of four GOKU-net variants across both reconstruction and forecast tasks. These variants encompass single and multiple shooting methods, either with or without the use of the attention mechanism. For the reconstruction task, we compared them against four baseline models: LSTM, Latent ODEs, ANODE, and a naïve predictor. In addition to these, we also included VAR in the set of baseline models for the forecast task.

All models were exclusively trained on the reconstruction task and their forecasts were generated during the evaluation stage. The prediction error was quantified using the normalized root mean square error (NRMSE) between the target ground truth and either the reconstruction or the forecast.

### 3.1 Simulated data

Figure 2a shows the performance of the different models in the reconstruction task on a synthetic dataset generated with the latent dynamics of three stochastic Stuart-Landau oscillators. In this context, the GOKU-net variants employing the multiple shooting method demonstrated significantly lower errors compared to other models. Notably, GOKU-nets that utilized the attention mechanism consistently showed better outcomes relative to their corresponding basic models. The single shooting variant particularly benefited from the attention mechanism. In the multiple shooting scenario, the performance boost due to attention was pronounced when using fewer training samples. However, when the number of training samples exceeded 600, the performance of both models converged to much closer values. However, GOKU-UI, which incorporates both the attention mechanism and multiple shooting, was still significantly the best-performing model. Wilcoxon signed rank tests, comparing GOKU-UI with each of the other models, for each number of training samples independently, yielded all p-values < 0.02 after Holm correction.

In contrast, the Latent ODEs, whose training instances consistently became trapped in local minima associated with the mean of the time series, markedly underperformed and exhibited errors akin to those of the naïve predictor. A similarly poor, though slightly better performance was obtained by ANODE. On the other hand, the LSTMs outperformed the basic GOKU-nets trained with the single shooting method. GOKU-UI, when trained on a mere 300 samples, yielded a reconstruction performance comparable to that achieved with 4800 samples. Moreover, when trained on just 150 unique samples, GOKU-UI outperformed all other single shooting models, even those trained on a dataset 32 times larger.

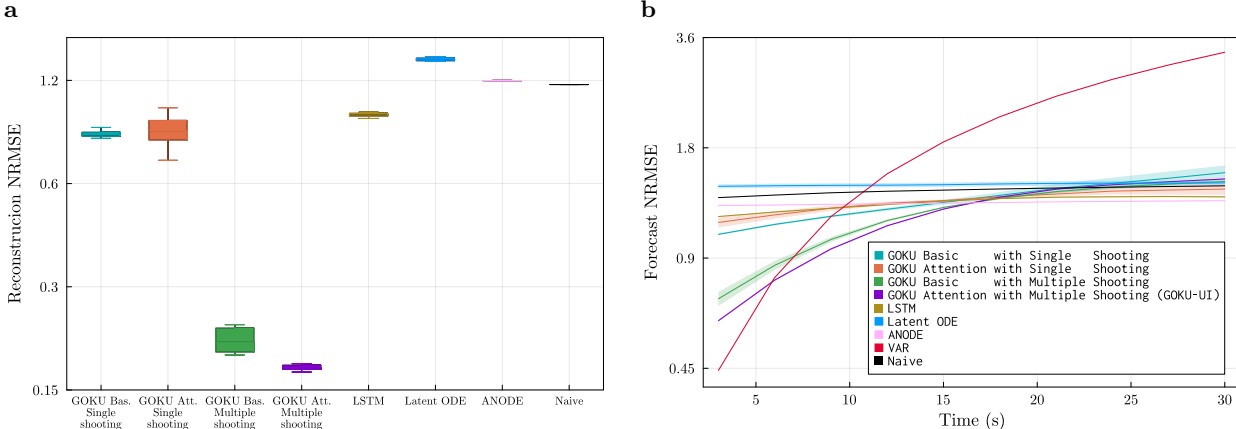

Figure 3: Comparison of the reconstruction (a) and forecast (b) performances on testing fMRI data, measured by the normalized RMSE. The averages were taken with respect to the input dimensions and time span, while the shaded regions correspond to standard errors derived from various training runs with multiple random seeds.

When forecasting 20 temporal points beyond the reconstruction limit, we observed a similar trend, as depicted in Figure 2b. However, in this task, GOKU-nets employing single shooting with the attention mechanism outperformed LSTMs. Except at the lowest end of the training data size, where VAR recorded the lowest forecasting error, GOKU-nets using multiple shooting consistently demonstrated superior overall performance. It is expected that simpler statistical methods, such as VAR, can sometimes surpass more complex machine learning models in scenarios with scarce training data. Nevertheless, GOKU-UI significantly outperformed other models when using 150 and 300 training samples (all p-values $< 0.05$, according to Wilcoxon signed-rank tests conducted independently for each sample size, after Holm correction). Beyond this, its performance was statistically indistinguishable from that of the basic GOKU model with multiple shooting (p-values $> 0.05$, Wilcoxon signed-rank tests, Holm corrected).

## 3.2 Empirical data

In a separate analysis, we trained GOKU-net models on human brain recordings, employing 20 coupled stochastic Stuart-Landau oscillators to govern the dynamics in their latent space. These models were trained alongside baseline models on 246 samples of 11-component ICA time series obtained from fMRI data. As shown in Figure 3a, unlike the simulated data scenario, the addition of the attention mechanism did not improve the reconstruction performance of the single shooting GOKU-net models. However, the multiple shooting training method again resulted in a significant improvement factor (p $< 0.001$, Wilcoxon signed-rank test). Remarkably, the GOKU-UI model, which integrates both multiple shooting and the attention mechanism, achieved a median reconstruction error that was five times lower than that of single shooting baseline models. Specifically, GOKU-UI also exhibited a significantly lower reconstruction NRMSE compared to the basic GOKU model that employed multiple shooting (p $< 0.04$, Wilcoxon signed-rank test). For a qualitative performance comparison between GOKU-UI and the original GOKU-net, refer to representative reconstruction plots in the Supplementary Information E.

Models trained exclusively for reconstruction were also assessed in a forecast task, which consisted of predicting the time series evolution immediately beyond the reconstruction limit. Figure 3b illustrates the performance trends of the different models as the forecast horizon extends. Notably, VAR exhibited the best short-term prediction accuracy, but its effectiveness diminished rapidly over time. This trend, in line with findings from Abrevaya et al. (2021), can be attributed to the relative slowness of the fMRI signal. The hemodynamic response function (HRF) in fMRI, which convolves underlying neural activity, typically operates over a timescale of about 10 seconds. Given the time resolution of fMRI is around 3 seconds, this temporal

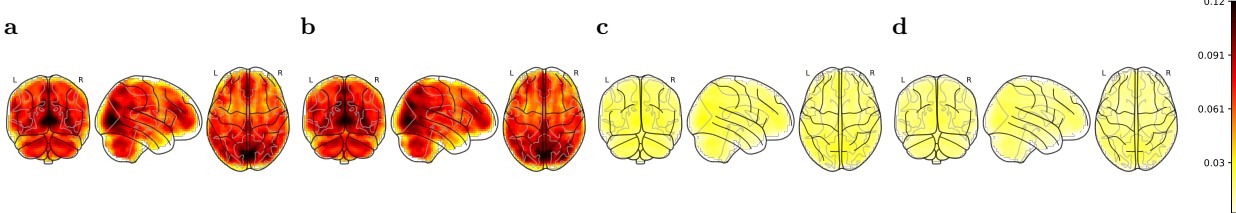

Figure 4: Comparison of the averaged errors in whole brain reconstructions from GOKU Basic with single shooting (a), GOKU Attention with single shooting (b), GOKU Basic with multiple shooting (c) and GOKU Attention with multiple shooting (d). The color scale represents the RMSE normalized by the maximum voxel value in the original brain data. The averages were calculated across all testing samples, time and random seeds.

setup implies that genuine signal fluctuations are unlikely to manifest within less than three to four time points. Furthermore, fMRI noise is characterized by strong autocorrelation, as noted in Arbabshirani et al. (2014). Additionally, during preprocessing, data undergo bandpass filtering within the range of 0.01-0.16 Hz, a process designed to enhance the neural signal-to-noise ratio while aligning with the HRF's natural filtering properties, but this also inherently limits the observation of rapid changes in the fMRI signal.

On the other hand, examining the performance of the GOKU-net models reveals that the multiple shooting method was again a key factor for improvement. Initially, GOKU-UI models tended to start their forecasts with a larger error margin than VAR but exhibited slower degradation, maintaining better performance than other models for up to 15 seconds. Beyond this point, GOKU-UI was outperformed by other baseline models, though it continued to exhibit marginally better performance compared to the basic GOKU-nets with single shooting. Nonetheless, all models demonstrated limited efficacy in long-range forecasts, a foreseeable outcome considering the complexity of the system under study. Given their exclusive training on reconstruction tasks, these models present opportunities for further refinement and enhancement.

In addition to evaluating performance on temporal ICA components, our study extends to reconstructing the full brain by linearly combining the components of ICA spatial support, each weighted by their corresponding time courses. Figure 4 demonstrates this approach by comparing whole brain reconstruction errors across the four GOKU-net variants. Each variant's performance is quantified using the RMSE, normalized by the maximum voxel value, and is visualized through a *glass brain* representation from three distinct perspectives. This comparison distinctly highlights the beneficial impact of the multiple shooting training approach in enhancing model performance. The visualization effectively conveys the quantitative differences in reconstruction errors and also offers an intuitive understanding of the spatial distribution of these errors across the brain. For additional insights, including a visual representation of the ICA spatial support and an analysis of whole brain forecasts, readers are directed to the Supplementary Information F.

## 4 Discussion and Limitations

Our research focused on two key enhancements to the baseline GOKU-net model: the addition of a basic attention mechanism and the implementation of a multiple shooting method. Independently, each of these modifications tended to improve the model's performance in both reconstruction and forecast tasks. The use of multiple shooting yielded the most substantial improvement. In particular, GOKU-UI, a composite of both enhancements, exhibited the best overall performance. During the evaluation on the synthetic dataset, GOKU-UI demonstrated remarkable efficiency with respect to the training data. With a mere 150 training samples, it outperformed all other single shooting baseline models, even when those had employed training sets 32 times larger (Figure 2). On average, for the synthetic dataset, GOKU-UI achieved a threefold reduction in reconstruction errors and a twofold reduction in forecast errors when compared to the original GOKU-net. In the case of the empirical dataset, compared to the original GOKU-nets, GOKU-UI managed to accurately capture the complex brain dynamics, achieving a fivefold reduction in reconstruction error. The GOKU-UI, while an extension of GOKU-nets rather than a completely new model, significantly enhances

performance by integrating attention and multiple shooting. This has resulted in large effect sizes across various applications, as detailed in Supplementary Tables 1 and 2. The notable exception is in fMRI future predictions beyond 15 seconds, where these enhancements are less effective.

Furthermore, we implemented GOKU-nets in the Julia Programming Language, which broadened the capabilities of the model. It helped overcome the initial limitation to Ordinary Differential Equations (ODEs) and facilitated the use of a wider range of differential equation classes (e.g., SDEs), as well as alternative advanced solvers and sensitivity algorithms. The unique SciML ecosystem of the Julia Language proved to be a potent and effective tool for research at the intersection of dynamical systems and machine learning.

Although the Stuart-Landau model has appeared in prior literature modeling brain dynamics (Deco et al., 2017; Donnelly-Kehoe et al., 2019; Ipiña et al., 2020), it has generally been employed with fixed coupling between oscillators. In these empirical studies, structural connectivity estimates were derived from Diffusion Tensor Imaging (DTI). Other parameters were adjusted to maximize some goodness-of-fit criterion between the simulated and empirical averaged functional connectivities. In contrast to those studies, the methodology employed in the current work targets the minimization of residuals between the simulated and empirical time series themselves. On the other hand, to our knowledge, this is the first instance of employing Stuart-Landau oscillators to model latent brain data dynamics while simultaneously inferring oscillator connectivity and learning the nonlinear transformation into the latent space. This approach differs from previous works, such as Abrevaya et al. (2021), which fitted the more general van der Pol oscillators to the latent representation of brain data. Specifically, their learning process involved two separate procedures: encoding in the latent space and parameter estimation of the differential equations. In contrast, our GOKU-UI model integrates these processes into a single end-to-end training regime. As discussed in Ramezanian-Panahi et al. (2022), this approach provides the unique advantage of integrating the known dynamics that govern the system, thus facilitating the interpretability and potential applicability with a smaller training set, while still maintaining the flexibility to learn nonlinear dependencies.

We have demonstrated these capabilities by training the GOKU-UI on fMRI data and encoding whole-brain dynamics into a latent representation. This representation's temporal evolution is effectively modeled by a low-dimensional, interpretable dynamical system, which can yield profound insights into brain functionality, such as the inference of functional connectivity. Beyond theoretical understanding, the model also holds potential for applied usage, including the classification of mental states or psychiatric conditions. These applications could either leverage the parameters of the differential equations or exploit higher-level features of the latent system, such as its attractor topology.

Despite its versatility as a tool in the SciML toolbox, GOKU-UI's main advantage may sometimes also be its primary limitation: unlike traditional, more agnostic machine learning models, GOKU-UI requires a preliminary differential equation model hypothesized to govern the data's intrinsic temporal dynamics. This requirement may be challenging to meet in many cases. For example, with latent ODEs, one can bypass this task, allowing another neural network to learn the differential equation. However, the significant complexity of the system under investigation, as evidenced by our experiments, could potentially hinder the efficacy of this method.

To the uninitiated in the field of dynamical systems, the process of proposing a specific differential equation to model the data's intrinsic but not immediately evident dynamics might seem like a guessing game. However, this approach has been the successful foundation of the field of physics since the era of Newton. The application of GOKU-UI to a new problem might not be as straightforward as a general-purpose black-box neural network model. Still, when guided by the vast theory of dynamical systems, it is not only possible, but potentially highly fruitful.

## Acknowledgments

Guillaume Dumas was supported by the Institute for Data Valorization, Montreal (IVADO; CF00137433 & PRF3) Professor Startup & Operational Funds, the Fonds de la Recherche en Santé du Québec (FRQ; 295289; 295291) Junior 1 salary award, the Natural Sciences and Engineering Research Council of Canada (NSERC; DGECR-2023-00089), the Brain Canada Foundation (2022 Future Leaders in Canadian Brain Research program), and the Azrieli Global Scholars Fellowship from the Canadian Institute for Advanced Research

(CIFAR) in the Brain, Mind, & Consciousness program. Irina Rish, Mahta Ramezanian Panahi and Jean-Christophe Gagnon-Audet acknowledge the support from the Canada CIFAR AI Chair Program, the Canada Excellence Research Chairs (CERC) program. Furthermore, Mahta Ramezanian Panahi acknowledges the UNIQUE Center support. Finally, the work of Silvina Ponce Dawson and Germán Abrevaya was funded by UBA (UBACyT 20020170100482BA) and ANPCyT (PICT-2018-02026, PICT-2021-I-A-00128, PICT-2021-III-A-00091). The computational resources used in this work were provided (in part) by the HPC center DIRAC, funded by Instituto de Física de Buenos Aires (UBA-CONICET) and part of the SNCAD-MinCyT initiative, Argentina.

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

## Supplementary Information

### A   Models architectures

Referring to the diagram in Figure 1, the specific architecture used for the different models, for both simulated and empirical data experiments, is as follows:

### A.1   Basic GOKU-nets

**Feature Extractor**

ResNet with 4 fully-connected layers, each with 200 neurons and using mish activation functions (Misra, 2019). Input dim = dimension of input data. Output dim = 128.

**Pattern Extractor**

Initial values path: an RNN with 2 layers and 64 neurons in each with ReLU activations. Input dim = 128. Output dim = 64.

Parameters path: Bidirectional LSTM with 2 layers and 64 neurons in each. Input dim = 128. Output dim = 128. Note that the dimension of the output of the forward LSTM and the backward LSTM are 64 but when concatenating them, the resulting output dimension is the given one.

**Latent in**

Initial values path: single-layered fully connected NN. Input dim = 64. Output dim = 64.

Parameters path: fully connected NN with 1 layer. Input dim = 128. Output dim = 128.

**Latent out**

Initial values path: fully connected NN with 2 layers and 200 neurons in the hidden layer, using no activation function (identity). Input dim = 64. Output dim = number of state variables of the differential equation.

Parameters path: fully connected NN with 2 layers and 200 neurons in the hidden layer, using sigmoid activation function. The parameters are projected from the interval [0, 1] to the desired range when integrating the differential equation. Input dim = 128. Output dim = number of parameters of the differential equation.

**Differential Equation layer**

The predefined differential equation is solved numerically for each of the sets of parameters and initial conditions provided by the previous layer. The output is the trajectories at time points equivalent to the input data.

**Reconstructor**

ResNet is similar to the one in the Feature Extractor, except that in this case the input dimension is the number of state variables of the differential equation and the output dimension is the one corresponding to the input data.

### A.2   GOKU-nets with attention

With the exception of the Pattern Extractor, the rest of the layers in the GOKU-nets with attention model remain identical to those in the basic GOKU-nets.

**Pattern Extractor**

Initial values path: LSTM with 1 layer. Input dim = 128. Output dim = 128.

Parameters path: Bidirectional LSTM (BiLSTM) with 1 layer. Input dim = 128. Output dim = 128. A fully connected NN with input and output dimensions of 128 is used for the attention mechanism. This

attention NN processes all the output sequences of the BiLSTM, after which a softmax is applied across the time dimension in order to obtain the attentional scores that will be used in the weighted sum of all the time steps returned by the BiLSTM.

### A.3 LSTM baseline model

The whole architecture is the same as in the basic GOKU-net, except for the Differential Equation layer, which is replaced by an LSTM:

**LSTM layer**

We used a single-layered LSTM with input and output dimensions set to $z\_dim$. This value is determined in each experiment to ensure that the total number of parameters in the LSTM model closely matches that of the corresponding GOKU-UI. For the simulated dataset experiments, we set $z\_dim = 42$. In the case of the empirical dataset experiments, $z\_dim = 105$. The LSTM operates recursively. It takes as its first input the value equivalent to the initial condition in differential equations. Subsequently, the model feeds back its last output as the new input, continuing this process until the number of time steps matches that of the model's input.

### A.4 Latent ODE baseline model

The whole architecture is the same as in the basic GOKU-net, except for the Differential Equation layer, which is replaced by a Neural ODE:

**Neural ODE layer**

Neural ODE is parametrized by a fully connected NN with 3 layers and $node\_hidden\_dim$ neurons in each. The input and output dimensions are given by $z\_dim$, which is the number of state variables. In the case of the simulated dataset experiments, the number of state variables was selected to match the true latent dimension $z\_dim = 6$ and the number of neurons in each layer was adjusted so that the total number of parameters in the model matched as closely as possible that of the corresponding GOKU-UI, resulting in $node\_hidden\_dim = 137$. On the other hand, in the case of the fMRI experiments, the number of state variables was set to $z\_dim = 20$ and $node\_hidden\_dim = 317$, also matching the total number of parameters of the corresponding GOKU-UI model.

### A.5 ANODE

In the case of the ANODE model, there is no latent representation, so it doesn't use the same Latent DE structure as the previous models. The neural ODE in this case is parametrized by 3 layers of fully connected NN, with their sizes adjusted to match the size of the GOKU-UI: 380 neurons in the cases of the simulated dataset and 675 neurons in the case of the empirical brain data. In both cases, the augmentation was of 100 dimensions.

### A.6 VAR

The VAR model is also applied directly to the input data without any latent representation inference. The implementation from the Python package `statsmodels` was used. The lag order was selected in each case as such that minimized the AIC in the training datasets.

## B  Comprehensive description of experiments

### B.1  Simulated dataset generation

The high-dimensional simulated dataset used for training the model was constructed based on the simulations of 3 coupled Stuart-Landau oscillators (Eqs. 2) with different random sets of parameters. Each set of parameters corresponds to a different training sample. Whenever we used the Stuart-Landau model in our experiments (both when generating the dataset and when using it inside the GOKU-nets), the time was

rescaled by multiplying the right-hand side of Eqs. 2 by 20. Thus, when integrating the equations with the used $dt = 0.05$, the input sequences of length 46 time steps contain a few oscillations. The parameters $a$, $\omega$ and $C$ were sampled from uniform distributions within the following ranges

$$a \in [-0.2, 0.2]; \quad \omega \in [0.08\pi, 0.14\pi]; \quad C \in [0, 0.2]$$

while $G = 0.1$ and $\eta = 0.02$. On the other hand, the initial conditions for the six state variables were sampled from uniform distributions within the ranges $[0.3, 0.4]$. For each set of parameters and initial conditions, the system is integrated with the SOSRI solver, a Stability-optimized adaptive strong order 1.5 and weak order 2.0 for diagonal/scalar Ito SDEs, from the DifferentialEquations.jl Julia package (Rackauckas & Nie, 2017). The complete time span of the integration is 35 units of time and the trajectories are saved every 0.05, resulting in 700 time points. The first 100 time steps are trimmed, in order to remove possible initial transients. Afterwards, a fixed random linear transformation is independently applied to each of the 600 remaining time steps, in order to obtain 784 dimensions. In other words, every state vector of length 6 from each sample is multiplied by the same $784 \times 6$ matrix, initialized by randomly sampling from a uniform distribution in the range [-1, 1]. A training dataset was created with 4800 samples, which serves as the source for the different training instances using various sizes of training sets, ranging from 75 to 4800 samples. Two other datasets were generated: one with 200 samples for validation and another with 900 samples for use in the subsequent model evaluations. A schematic representation of the data splits is presented in the left pane of Figure 5. Keep in mind that each sample corresponded to a different set of initial conditions and parameters of the differential equations, ensuring that the three datasets are different.

## B.2 Empirical dataset generation

We used resting state fMRI data from 153 participants, obtained from the Track-On HD study (Klöppel et al., 2015). The data underwent pre-processing, as described in Polosecki et al. (2020), and a 20-component Canonical ICA (Varoquaux et al., 2010) was performed. Upon inspecting the resulting 20 components, 9 were identified as artifacts and thus discarded, leaving 11 components for further analysis in our experiments. Each subject contributed data from two visits, accumulating a total of 306 data samples. Each sample comprised 160 time points, obtained at a temporal resolution of 3 seconds.

In contrast to the simulated data scenario, the empirical data was considerably limited. Consequently, we optimized our data splitting approach. To avoid 'double-dipping', we ensured that the training and testing datasets comprised entirely distinct samples. However, for the training and validation sets, we utilized the same samples, implementing a temporal split. Specifically, approximately 20% of the data samples (n=60) were reserved for testing, while ensuring a balanced representation of the sex, condition, and measurement site. The rest of the data samples (n=246) were allocated for training and validation purposes. Within this subset, the initial 114 time points were designated for model training. The remaining data served for validation and for an criterion early training termination. Finally, the training, validation, and test splits were all normalized by the standard deviation of the training set. The right pane of Figure 5 displays a diagram of the data partition employed for the fMRI dataset.

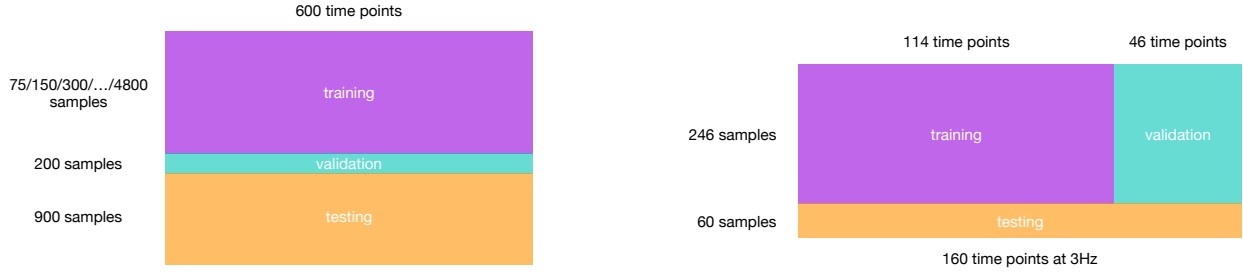

Figure 5: Graphical representation of the data splits of the Stuart-Landau synthetic dataset (left pane) and fMRI dataset (right pane) into distinct sets for training, validation, and testing ensures evaluations of models on unseen data during the training phase.

### B.3 Training settings

All the experiments underwent the same training procedure with identical hyperparameters, which will be described here.

The input sequence length for all the models was 46 time steps, and the batch size was set at 64. As described above, the full length of each sample in the training sets was 600 time steps for the synthetic dataset and 114 for the fMRI dataset. The procedure for generating a batch of training data is as follows: First, 64 samples that have not been used previously in the current training epoch are randomly selected. Then, for each sample, a 46 time-step-long interval is randomly chosen within the 600 or 114 time steps available in the full training sample length. Refer to Figure 5 for a visual representation of the data partitioning. The GOKU-net based models, contain the same Stuart-Landau differential equations as described above, however, the allowed ranges of parameters differ from the ones used during the generation of the synthetic dataset. In order to be closer to a real world use-case we allow for a wider range of parameters than those actually used for generating the data, since in principle one would not know the true range:

$$a \in [-1, 1]; \quad \omega \in [0, 1]$$

while keeping, the other parameters the same except of the connectivity in the empirical fMRI training, in which case it was allowed to be negative: $C \in [-0.2, 0.2]$. The differential equations definitions were optimized for higher computational performance with the help of ModelingToolkit.jl (Ma et al., 2021b). During training, they were solved with the `SOSRI` solver, a Stability-optimized adaptive strong order 1.5 and weak order 2.0 for diagonal/scalar Ito SDEs, from the DifferentialEquations.jl Julia package (Rackauckas & Nie, 2017). The sensitivity algorithm used was `ForwardDiffSensitivity` from the SciMLSensitivity.jl package (Rackauckas et al., 2020). The models were defined and trained within the deep learning framework of the Flux.jl package (Innes et al., 2018). The experiments were managed using DrWatson.jl package (Datseris et al., 2020).

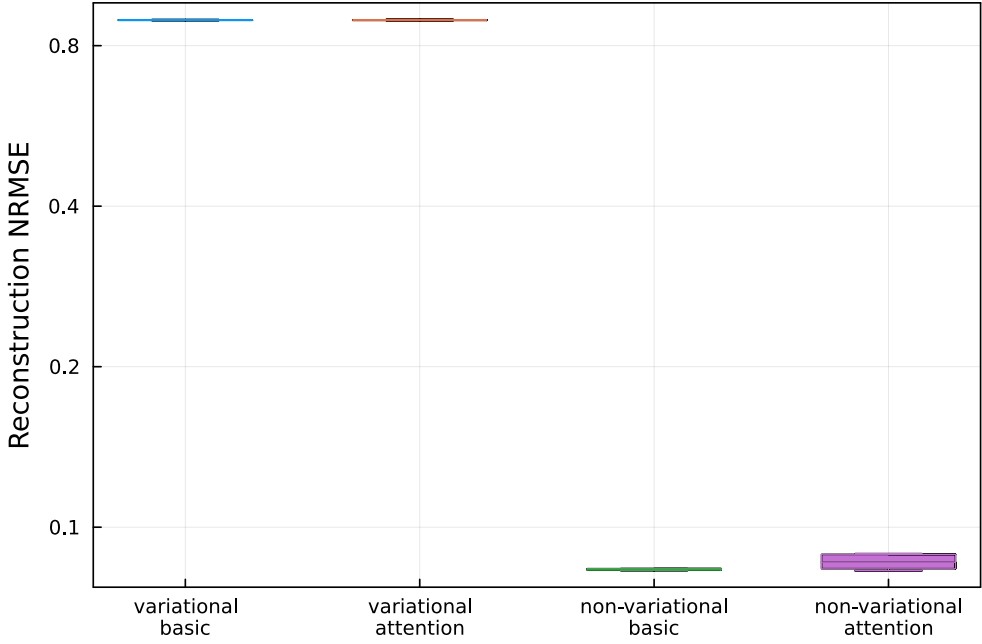

Figure 6: Box plot of the normalized RMSE on the test synthetic dataset for GOKU-nets trained with multiple shooting either in their basic or attention variants and in a variational or non-variational version. All hyperparameters are kept as in the main paper experiments.

The model was trained with Adam with a weight decay of $10^{-10}$, and the learning rate was dynamically determined by the following schedule. The learning rate begins with a linear growth (also referred to as

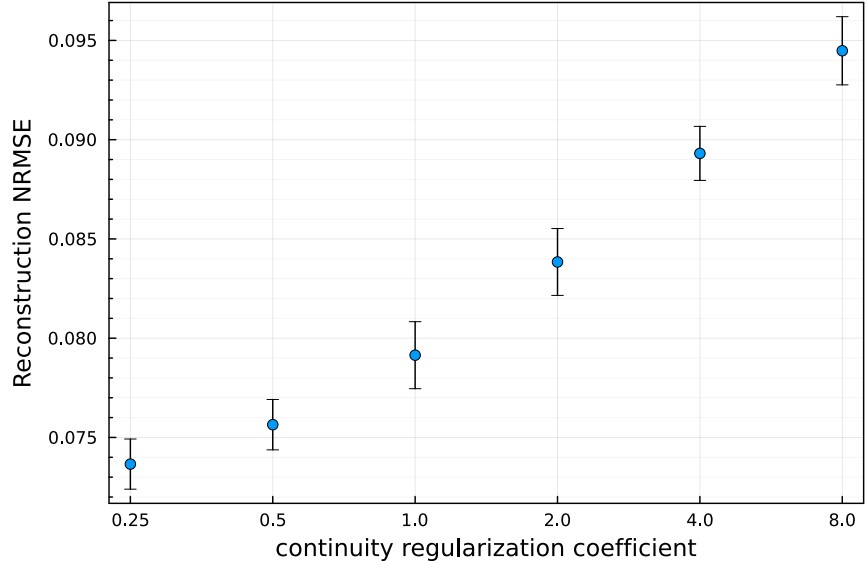

Figure 7: Reconstruction normalized RMSE of GOKU-UI on the test synthetic dataset for different values of the regularization coefficient in the loss function regarding the continuity constraint. All other hyperparameters are kept as in the main paper experiments.

*learning rate warm-up*) from $10^{-7}$, escalating up to 0.005251 across 20 epochs. Afterwards, it maintains that value until the validation loss stagnates (has not achieved a lower value for 50 epochs), at which point it starts a sinusoidal schedule with an exponentially decreasing amplitude.

For the multiple shooting training, all the presented experiments used a time window length of 10, therefore partitioning 46-time-steps-long sequences into 5 windows with their endpoints overlapping. The regularization coefficient in the loss function for the continuity constraint had a value of 2.

Since we found that models with variational autoencoders underperform their non-variational versions (see Figure 6), all the results presented in this work were obtained using non-variational GOKU-nets. This is, instead of sampling from normal distributions in the latent space as depicted in Figure 1, we pass forward the mean values $\mu_{z_0}$ and $\mu_\theta$. Thus, the associated loss function does not have the KL divergence term associated with the ELBO but retains the reconstruction loss given by the mean squared error between the output of the model and the input, normalized by the mean absolute value of the input. In addition, when multiple shooting training is employed, the extra term regarding the continuity constraint is included in the loss function. This extra term consists of the mean squared differences between the last point of a window and the initial from the next one, divided by the number of junctions and multiplied by a regularization coefficient. Please, note that this continuity regularization is performed in the state space of the differential equation and not in the input space.

## C Further exploration of the model

In Figure 6, a box plot is presented, comparing the reconstruction performances on the synthetic dataset when the models are variational versus when they are not.

Figures 7 and 8 present the performance of the GOKU-UI on the synthetic dataset when trained using different hyperparameters. As a reminder, the base values used in our experiments for the continuity regularization coefficient, window length, and input sequence length are 2, 10 and 46 respectively. In Figure 7 all hyperparameters are kept the same, except for the continuity coefficient. Similarly, Figure 8 shows variations with respect to the window length and Figure 8, respect to the input sequence length.

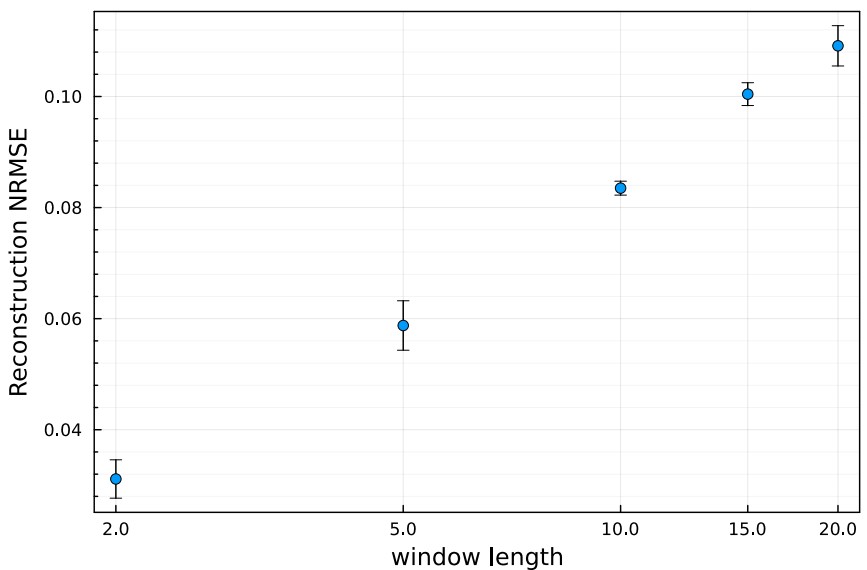

Figure 8: Reconstruction normalized RMSE of GOKU-UI on the test synthetic dataset for different values of time window length for the multiple shooting partition. All other hyperparameters are kept as in the main paper experiments.

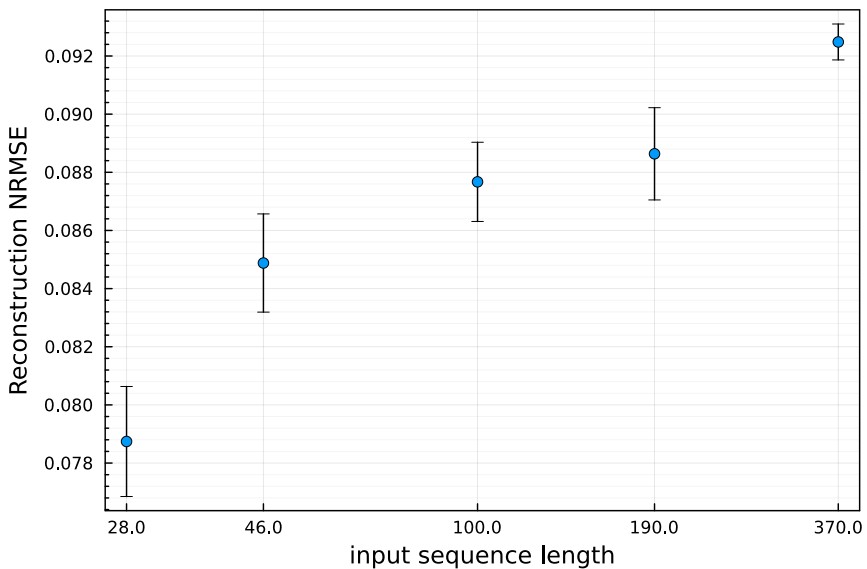

Figure 9: Reconstruction normalized RMSE of GOKU-UI on the test synthetic dataset for different values of time input sequence length. All other hyperparameters are kept as in the main paper experiments.

As evidenced by these results, there exist other sets of hyperparameters that produce higher performances than those obtained with the base hyperparameters used in our experiments. The set of hyperparameters had been selected based on a grid search but while using a different dynamical system. In any case, the results presented in this paper serve as a proof of concept, showing that even when using a sub-optimal set of hyperparameters, GOKU-UI demonstrates significantly better reconstruction and forecast performances with respect to the baselines models and other GOKU-nets variants tested.

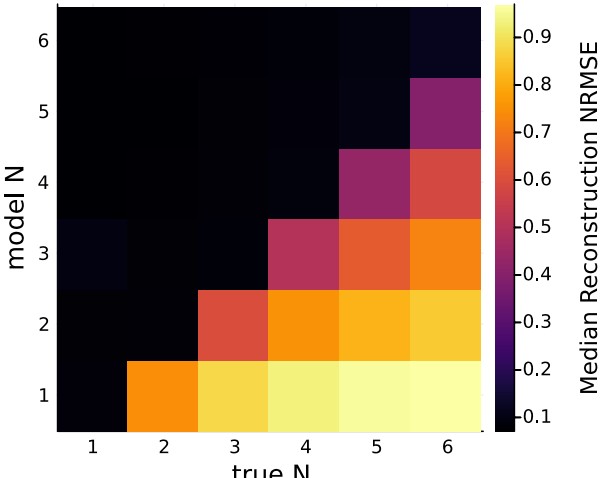

Figure 10: Reconstruction errors of GOKU-UI using different number of oscillators (*model N*) on synthetic Stuart-Landau datasets constructed with *true N* number of oscillators.

Motivated by the question of whether it would be possible to identify the latent dimensionality of some data using the GOKU-UI model, the following experiments were performed. GOKU-UI models with different numbers of Stuart-Landau oscillators were trained on synthetic datasets generated from distinct numbers of latent oscillators but all with the same input size of 784, so the latent dimensionality was not evident. Furthermore, as in the training settings from our other experiments, the allowed parameter ranges were wider in the GOKU-UI than when generating the datasets. The results of such experiments are presented in Figure 10, where color represents testing reconstruction errors for GOKU-UIs with *model N* oscillators on datasets built with latent *true N* oscillators. We see that with an increasing number of oscillators in the model, the error progressively diminishes until reaching the true number of latent oscillators in the data,

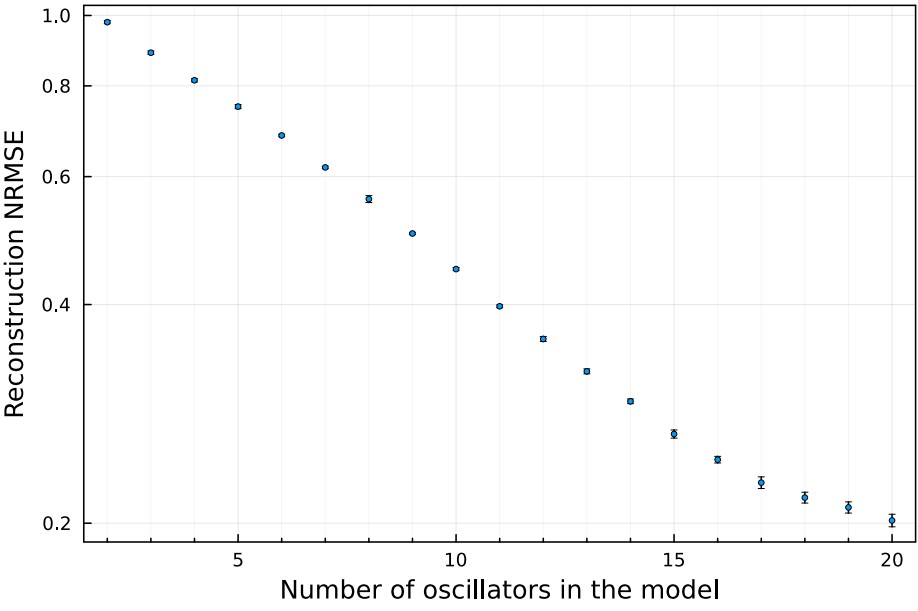

Figure 11: Reconstruction normalized RMSE on fMRI test data, while changing the number of Stuart Landau oscillators inside the GOKU-UI model.

and from there, the error gets abruptly reduced. Notably, when using more oscillators in the model than the true latent in the data, the model still learns equally well how to reconstruct it. However, the most salient feature is that in this simulated scenario, it is possible to identify the true latent dimensionality of the data as such of the number of oscillators in GOKU-UI for which the reconstruction error gets abruptly reduced.

A similar attempt was made to infer the latent dimensionality of the fMRI data. However, as seen in Figure 11(where the y-axis is in logarithmic scale), there is no abrupt descent in the reconstruction error for any number of oscillators inside GOKU-UI that we tried. Nevertheless, an inflection point can be identified for $N = 17$. We concluded that any $N \geq 17$ was adequate to use and chose $N = 20$ to perform the experiments presented in this study.

## D    Effect sizes

In this section we present the Cohen's d effect sizes comparing the performances (measured in RMSE) from original GOKU-nets (without attention and trained with single shooting) against the GOKU-UI. The effect sizes regarding the evaluations of the simulated Stuart-Landau experiments can be found in Table 1, while the corresponding for the empirical fMRI in Table 2. From these effect sizes, we can conclude that the improvements due to the incorporation of attention and multiple shooting into GOKU-nets are large (d > 0.8, according to the criterion suggested by Cohen (2013)) in all cases except for fMRI future predictions beyond 15 seconds.

| Training Samples | Cohen's $d$ Effect Size | |
|---|---|---|
| | Reconstruction Task | Forecast Task |
| 75 | 16.79 [11.51, 22.07] | 6.38 [4.22, 8.54] |
| 150 | 7.8 [5.23, 10.37] | 8.75 [5.9, 11.6] |
| 300 | 7.82 [5.24, 10.4] | 6.16 [4.06, 8.26] |
| 600 | 4.75 [3.04, 6.47] | 4.08 [2.54, 5.62] |
| 1200 | 5.52 [3.6, 7.45] | 5.26 [3.41, 7.12] |
| 2400 | 3.29 [1.94, 4.63] | 2.91 [1.65, 4.16] |
| 4800 | 3.86 [2.38, 5.35] | 3.27 [1.93, 4.61] |

Table 1: Cohen's $d$ effect sizes with 95% confidence intervals for testing performance comparison, measured in RMSE, between the original GOKU-net and GOKU-UI. This table corresponds to the experiments on simulated Stuart-Landau oscillators for both reconstruction and forecast tasks, and the effect sizes are calculated independently for each size of the training set.

| Training Samples | Cohen's $d$ Effect Size |
|---|---|
| | Reconstruction Task |
| 246 | 59.18 [37.98, 80.38] |

| Time (s) | Cohen's d Effect Size |
|---|---|
| | Forecast Task |
| 3 | 32.01 [20.51, 43.5] |
| 6 | 19.91 [12.71, 27.11] |
| 9 | 10.4 [6.54, 14.26] |
| 12 | 4.57 [2.64, 6.49] |
| 15 | 1.44 [0.3, 2.58] |
| 18 | 0.41 [-0.61, 1.44] |
| 21 | 0.31 [-0.72, 1.33] |
| 24 | 0.48 [-0.55, 1.51] |
| 27 | 0.72 [-0.32, 1.77] |
| 30 | 0.97 [-0.1, 2.04] |

Table 2: Cohen's $d$ effect sizes with 95% confidence intervals for testing performance comparison, measured in RMSE, between the original GOKU-net and GOKU-UI for empirical fMRI data. The table presents results for both reconstruction and forecast tasks, with effect sizes calculated independently for each forecast horizon in the latter.

## E    Reconstruction plots

To provide a visual representation of the model's performance, this section presents trajectories from both the synthetic and empirical fMRI test sets, along with their corresponding reconstructions by GOKU-UI and the original GOKU-nets (lacking attention mechanisms and trained with single shooting). The x-axis represents time steps in all cases. To display representative cases, samples were selected based on their mean reconstruction RMSE being closest to the median error across all samples. For the synthetic data, 11 components were randomly selected for display in Figures 12 and 13, due to the impracticality of displaying all 784 components. Each figure displays results from different instances of models, all trained with 4800 samples but each initialized with a unique random seed. For the fMRI data, all 11 ICA components are displayed in Figures 14 and 15.

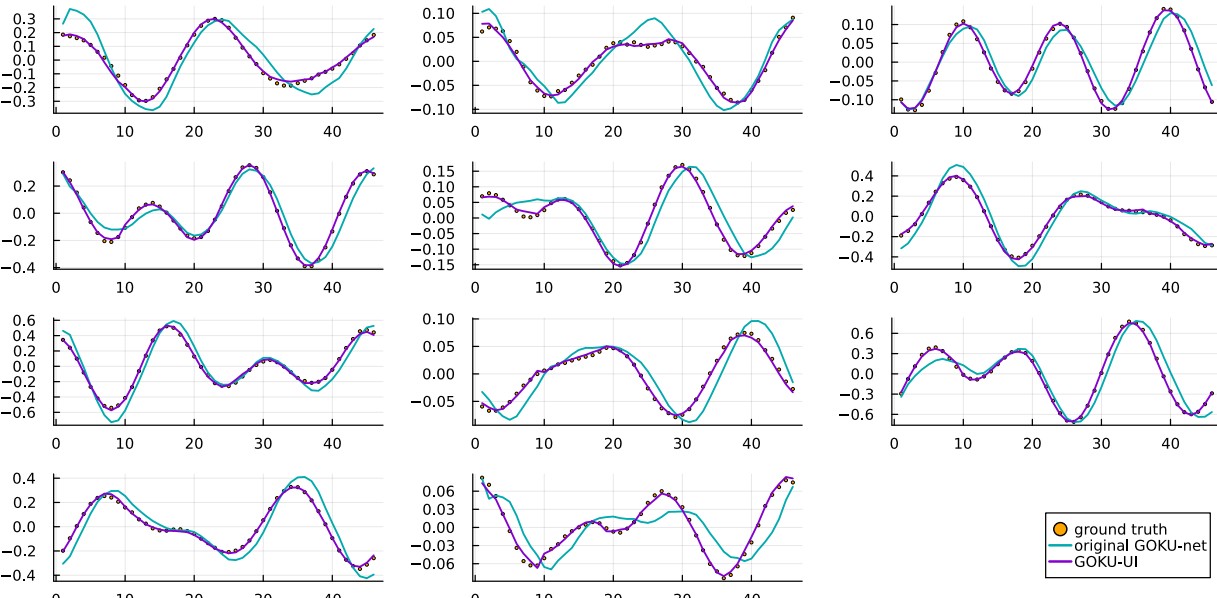

Figure 12: Representative example of a 46-time-step input sequence from the synthetic test set, accompanied by its reconstructions from both GOKU-UI and the original GOKU-nets (lacking attention mechanisms and trained with single shooting). The sample was selected so that its RMSE was the closest to the median error across all samples. 11 randomly selected components out of the 784 are displayed.

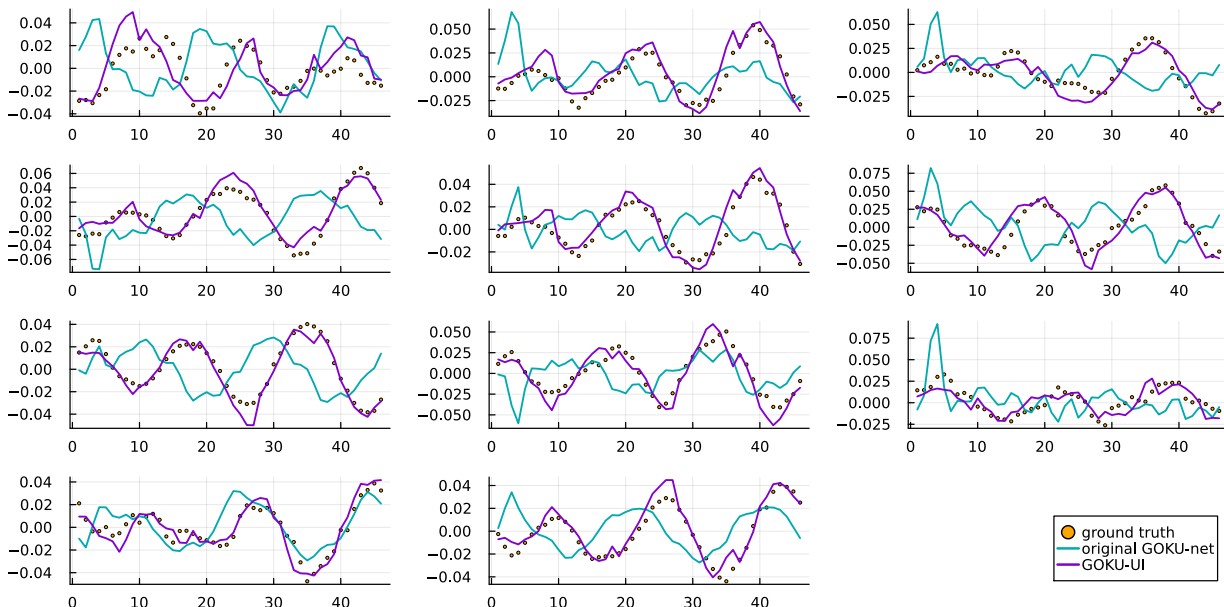

Figure 13: Representative example of a 46-time-step input sequence from the synthetic test set, accompanied by its reconstructions from both GOKU-UI and the original GOKU-nets (lacking attention mechanisms and trained with single shooting). The sample was selected so that its RMSE was the closest to the median error across all samples. 11 randomly selected components out of the 784 are displayed. This figure is similar to the previous one but presents results from different instances of the trained models, each initialized with a unique random seed.

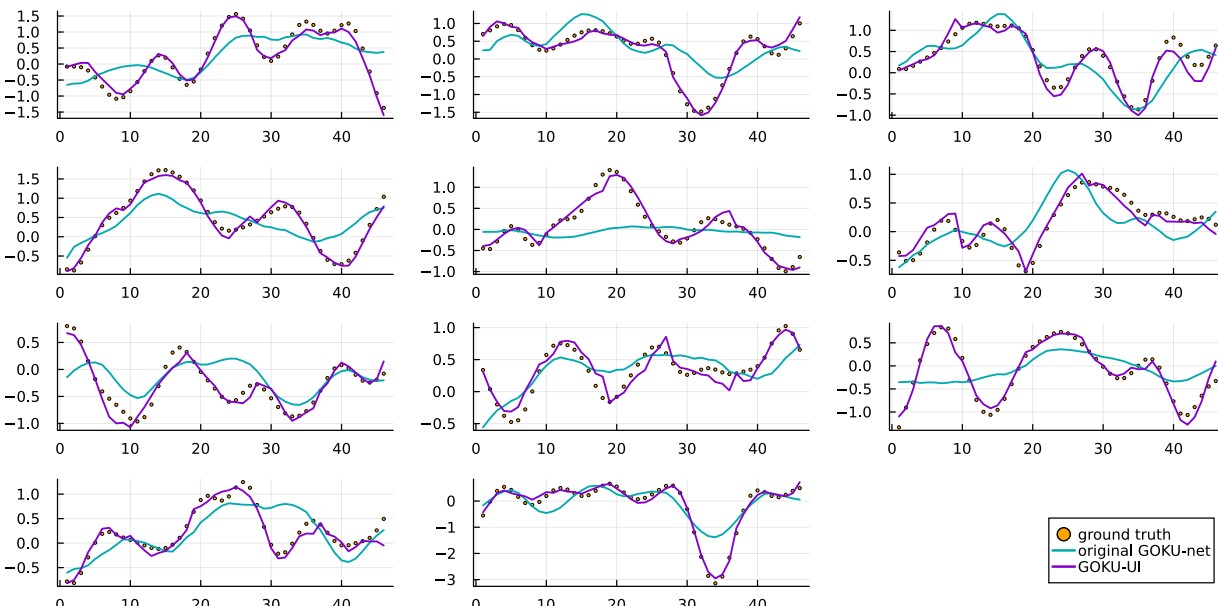

Figure 14: Representative example of a 46-time-steps input sequence for all considered ICA components from the empirical fMRI test set, accompanied by its reconstructions from both GOKU-UI and the original GOKU-nets (lacking attention mechanisms and trained with single shooting). The sample was selected so that its RMSE was closest to the median error across all samples. The x-axis represents time steps, each corresponding to 3 seconds.

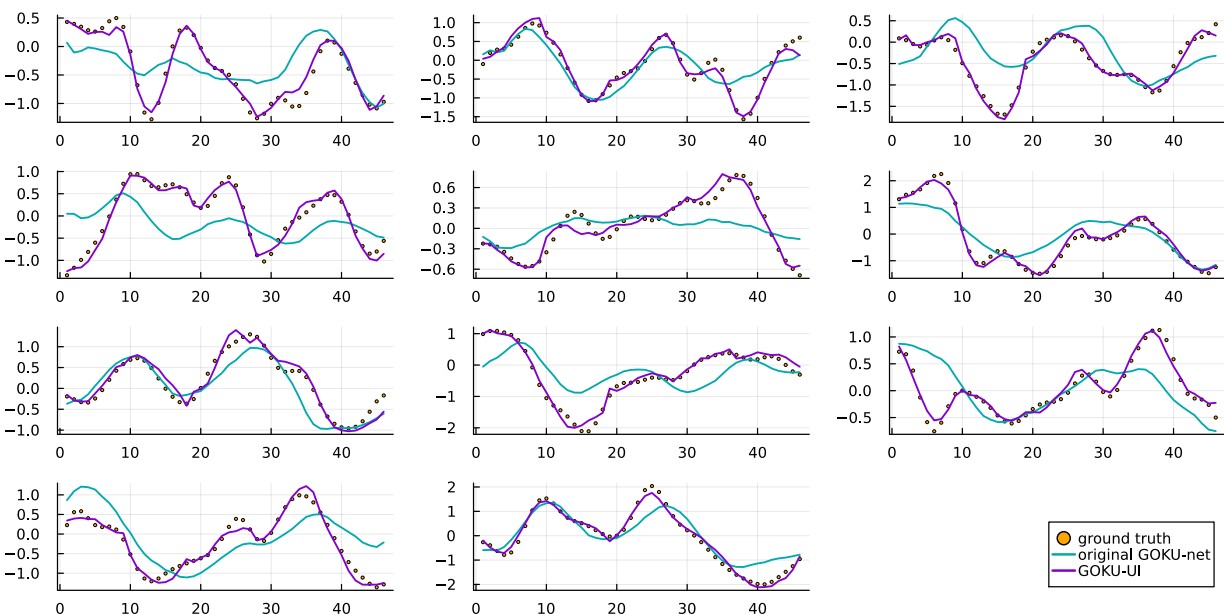

Figure 15: Representative example of a 46-time-steps input sequence for all considered ICA components from the empirical fMRI test set, accompanied by its reconstructions from both GOKU-UI and the original GOKU-nets (lacking attention mechanisms and trained with single shooting). The sample was selected so that its RMSE was closest to the median error across all samples. This figure is similar to the previous one but presents results from different instances of the trained models, each initialized with a unique random seed. The x-axis represents time steps, each corresponding to 3 seconds.

## F ICA spatial support and whole brain forecasts

The spatial support of the Canonical ICA used in the current study is shown in Figure 16. Since each spatial component extends across the whole brain, each component is displayed in a glass brain representation, from three different perspectives.

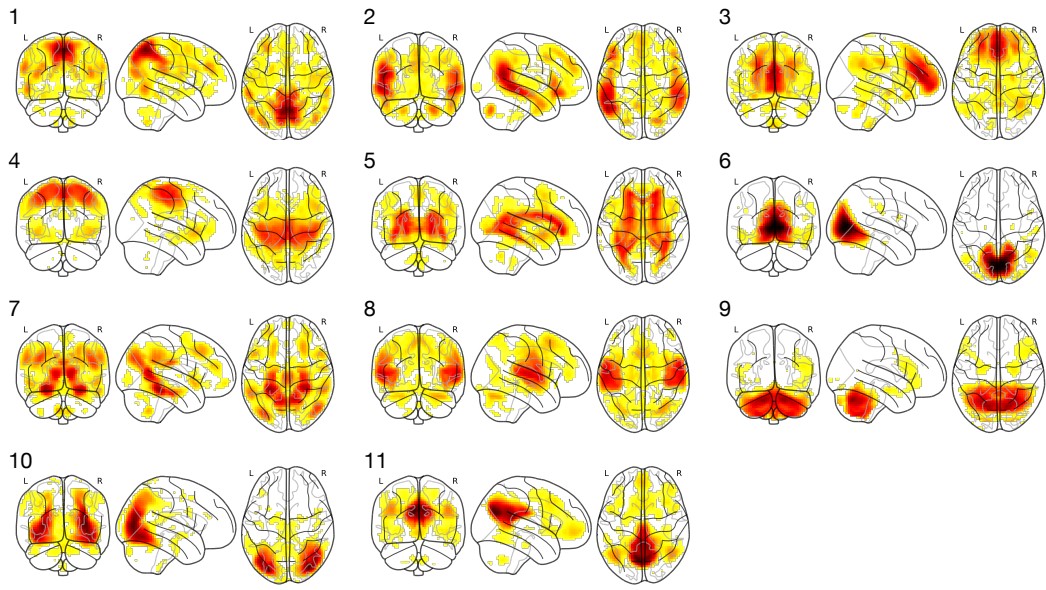

Figure 16: Spatial support of the Canonical ICA analysis employed in this study.

In Figure 17 a comparison of the error in future brain activity forecast between the original GOKU-net and GOKU-UI is presented. Color code represents the RMSE between the actual brain activity and the forecasts beyond the reconstruction limit generated by these models. In this full brain visual representation it is also noticeable that GOKU-UI generates better forecasts than its original GOKU-net counterpart. However, the deterioration of the forecast across the brain is not homogeneous in either case. Actually, both in the forecasts and reconstructions (see Figure 4) we notice that an accentuated error tends to be present in the region of precuneus and posterior cingulate cortex. This area is a critical hub of the default mode network and is particularly associated with reflective self-conscious thought and internally focused cognitive activities (Whitfield-Gabrieli et al., 2011). In this sense, the neural dynamics of the precuneus and posterior cingulate cortex region are less trivially entrained by the environment because they play a central role in the trafficking of information between all the other regions of the brain (Donnelly-Kehoe et al., 2019). As a consequence, they are key parts of the "dynamical core" of the brain, making their activity particularly difficult to predict, especially during resting state (Deco et al., 2017).

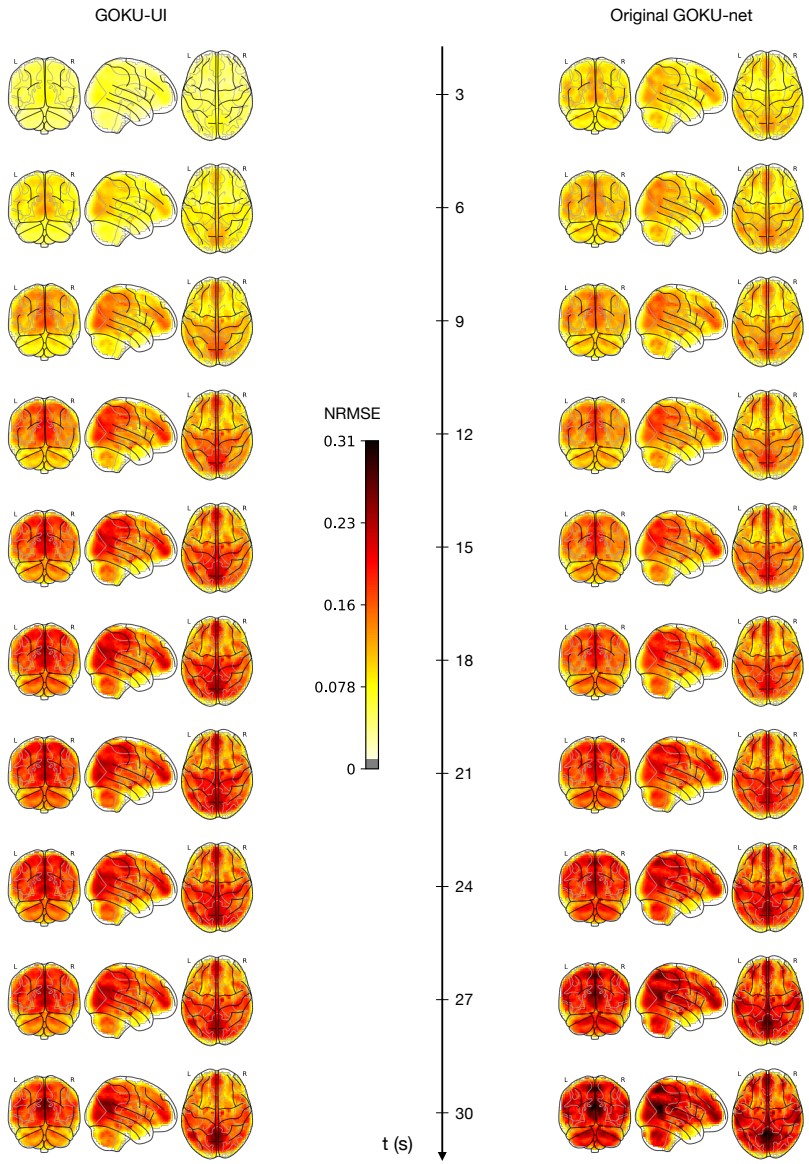

Figure 17: Time evolution of the whole brain activity, comparing the forecasts from the original GOKU-net respect to GOKU-UI. The color represent the RMSE normalized by the maximum voxel value and averaged over all testing samples and training instances associated to different random seeds.

