# OpenReview forum: "Effective Latent Differential Equation Models via Attention and Multiple Shooting"
_TMLR — Accepted by TMLR_

### Review · Reviewer_CTWL · 2023-10-30

**Summary Of Contributions:**

This paper offers two amendments to the original Generative ODE Modeling with Known Unknowns (GOKU) model. First, it introduces changes to the model's structure by substituting the final biLSTM elements in the Pattern Extractor with an attention weighted blend of all outputs for each sequence element. Second, it modifies the training process by adopting a 'multiple shooting' approach to sequence generation. The effectiveness of these alterations is evaluated against the original GOKU model and other alternatives, utilizing both simulated data with established ground truth and an fMRI dataset.

**Audience:**

Yes

**Broader Impact Concerns:**

no concerns

**Claims And Evidence:**

No

**Requested Changes:**

1.  Redo experiments reported in Figures 2 and 3 using a proper cross validation, when time-series are not split in parts and then distributed across training and test data. No part of any training time series should be used in the test.
2.  IMO, to ensure that the entire class of presented models isn't underperforming on the tasks demonstrated, the paper **must** include other models for baseline reference, such as the following [package for forecasting models](https://github.com/Nixtla/statsforecast/tree/main)
3.  Minor editorial problems I have noticed:
    -   Section 2.3.2 is called Empiric data. You must mean empirical evaluation?
    -   It would greatly enhance the accessibility of the referenced literature, at which the paper excels, if all the citations in the PDF were turned into clickable hyperlinks.
    -   First line of section 2.1 "could be thought as" -> "could be thought of as". Consider rewriting this sentence entirely - it is awkwardly phrased.

**Strengths And Weaknesses:**

# Strengths

1.  The paper begins with a well-structured introduction that seamlessly integrates both contemporary and historical references, providing a comprehensive overview of the field.
2.  The authors have done an excellent job in presenting their research; the paper at large is well-articulated and easy to understand.
3.  The approach to empirical evaluation is commendable, the authors begin with simulations and then transition into real data application. This succession is usually desirable when one is working with neuroimaging data where the ground truth is unknown.

# Weaknesses

1.  The empirical evaluation as it stands is unacceptable.
    1.  The key issue lies in evaluating the reconstruction while partitioning the identical time sequences (or subjects in the fMRI example) between the train and test sets. This 'double-dipping' results in overly optimistic results, as the model and ODE parameters are being optimized based on the very same time series data. **Please, separate the training and test subjects as well as simulated sequences.** I may be missing the point here and the propose model is literally an approach to fit an ODE to a specific dataset, but then the paper is putting forward quite different claims.
    2.  While the forecasting experiments may not be as vulnerable to the previous issues, the paper lacks a thorough evaluation of its forecasting performance.  The models that the paper does compare with are closely related to the type of model utilized. Meanwhile, time series forecasting is a vibrant field complete with numerous benchmarks. However, no comparison with the more standard forecasting models is provided.
2.  The paper lacks a convincing justification of the fMRI forecasting experiments. The use-case of this exercise neither explained nor explored. I believe a reader would appreciate more motivation in this section.

---

> ### Author Response · Authors · 2023-12-13
> **Response**
>
> Dear Reviewer CTWL,
>
> We sincerely appreciate your thorough review, which has been instrumental in enhancing the quality of our paper.
>
> 1. We apologize if our presentation was not sufficiently clear, but we would like to clarify that our experiments employed completely distinct training and testing sets, ensuring there was no 'double-dipping'. We have improved the clarity of our dataset generation and partitioning explanation with the following additions on page 7:
>
>   - In Section 2.3.1 Simulated Data:
> >Each sample corresponds to a different set of initial conditions and parameters for the SL differential equations. A validation set of another 200 samples was used for the training termination criteria, and a separate testing set containing 900 different samples was employed for the evaluation.
>
> - In Section 2.3.2 Empirical Evaluation:
> > The contribution of each subject was captured in two separate visits, yielding an aggregate of 306 data samples. These samples, with 160 time points each, were acquired at a temporal resolution of 3 seconds. In contrast to the simulated data scenario, the empirical data was considerably limited. Consequently, we optimized our data splitting approach. To avoid 'double-dipping', we ensured that the training and testing datasets comprised entirely distinct samples. However, for the training and validation sets, we utilized the same samples, implementing a temporal split. Specifically, approximately 20% of the data samples (n=60) were reserved for testing, while ensuring a balanced representation of the sex, condition, and measurement site. The rest of the data samples (n=246) were allocated for training and validation purposes. Within this subset, the initial 114 time points were designated for model training. The remaining data served for validation and for an criterion early training termination. For a visual representation of the dataset partitioning, please refer to Figure 5 in the Supplementary Information.
>
> - Furthermore, as indicated, to dispel any doubts about our data partitioning, we have added visual representations of the dataset partitions in the Supplementary Information (Figure 5, page 17).
>
> &nbsp;
>
> 2. We also thank you for the suggestion to include more classic statistical models as baselines for forecasting tasks. We explored Nixtla, the Python library you kindly recommended, but it turned out that all models therein are only for univariate time series, whereas our study involves multivariate time series. After consulting with the developers of Nixtla and confirming its limitation to univariate time series, we decided to include the Vector Autoregressive (VAR) model, a standard statistical method for forecasting multivariate time series. This decision was influenced by our prior experience with the VAR model in other studies and coincidentally was also the suggestion from Nixtla's developers. Consequently, we have included the VAR model as an additional baseline, along with the Augmented Neural ODEs model, following a request from another reviewer. All pertinent updates have been made to the manuscript, including revisions in the methods, results analysis, and associated discussions to accurately reflect these changes.
>
> &nbsp;
>
> 3. Thank you for pointing out the editorial issues. We have addressed each of them:
>
> - We corrected “Empiric data” to “Empirical Evaluation”.
> - We have made all citations in the PDF clickable for enhanced accessibility.
> - We rewritten the awkwardly phrased sentence in Section 2.1 to:
> > GOKU-nets can be considered a specific instance within the broader category of models we denominate Latent Differential Equation models (Latent DEs).
>
> &nbsp;
>
> 4.  Additionally, we acknowledge your point regarding the lack of a convincing justification for the fMRI forecasting experiments. In response, we have added the following text at the end of the first paragraph of Section 2.1 (page 3):
>
> > After encoding the data into its latent space, beyond simply generating an output that corresponds to the time span of the input, these models can integrate the differential equation further in time, allowing them to forecast the system's future evolution. This capability is not only crucial for validating a model, as is standard in scientific modeling, but also has diverse practical applications. These applications range from finance and weather forecasting to neuromodulation in the context of brain data, where accurate short-term forecasts can help compensate for delays in closed-loop feedback systems (Parastarfeizabadi & Kouzani, 2017).
>
> &nbsp;
>
>
> Thank you once again for your dedication and valuable insights in reviewing our paper. We greatly appreciate the time you have taken to provide constructive feedback.
>
> Sincerely,
>
> The Authors

---

> > ### Comment · Reviewer_CTWL · 2024-01-26
> > **my concerns were addressed**
> >
> > Dear authors, thank you for your patience.
> > Your response addresses my initial concerns.

---

### Review · Reviewer_ZBuz · 2023-11-04

**Summary Of Contributions:**

The proposed method extends GOKU-nets with the multiple shooting method
improvind forecast of non-linear dynamical system. The method is evaluated
on simulated Stuart-Landau (SL) oscillators and on fMRI rest data showing a gain in
reconstruction and forecast scores. Results show that the proposed is more
stable to train and is able to learn with moderate amount of data.

**Audience:**

Yes

**Broader Impact Concerns:**

The paper is overal well written with convincing experiments and literature
review, it is however not very amenable to a large audience not already
familiar with neural ODEs and GOKU-like methods.

**Claims And Evidence:**

Yes

**Requested Changes:**

Results on fMRI data are potentially interesting yet results do not
offer any insights on the data and neuroscience interpretation of the
results (eg where in the brain does the model perform better? etc.)
The results in appendix D show very good reconstruction plots but in
ICA space and we do not know what components look like etc.

**Strengths And Weaknesses:**

The paper is overal well written with convincing experiments and literature
review, it is however not very amenable to a large audience not already
familiar with neural ODEs and GOKU-like methods. Indeed the paper is
very short and comes without background to make the contribution more
approachable by people interested in the application of these methods.

---

> ### Author Response · Authors · 2023-12-13
> **Response (1 of N)**
>
> Dear Reviewer ZBuz,
>
> Thank you for your insightful feedback on our manuscript. We highly appreciate your constructive critique, particularly regarding the interpretability of our results in the context of neuroscience. Based on your suggestions, we have made several enhancements to our paper to address these concerns.
>
> 1. Whole Brain Reconstructions Errors Visualization: To address your point about the lack of interpretative insights in the fMRI data (eg where in the brain does the model perform better?), we have expanded Section 3.2. We now include a visualization of reconstruction errors for testing samples across the entire brain:
>
> > "In addition to evaluating performance on temporal ICA components, our study extends to reconstructing the full brain by linearly combining the ICA spatial support, weighted by their corresponding time courses. Figure 4 demonstrates this approach by comparing whole brain reconstruction errors across the four GOKU-net variants. Each variant's performance is quantified using the RMSE, normalized by the maximum voxel value, and is visualized through a 'glass brain' representation from three distinct perspectives. This comparison distinctly highlights the beneficial impact of the multiple shooting training approach in enhancing model performance. The visualization effectively conveys the quantitative differences in reconstruction errors and also offers an intuitive understanding of the spatial distribution of these errors across the brain. For additional insights, including a visual representation of the ICA spatial support and an analysis of whole brain forecasts, readers are directed to the Supplementary Information E."
>
> 2. Supplementary Information Enhancement: Following your suggestion, we have also enriched the Supplementary Information Section F. It now includes:
>
> - Visual representations of the ICA spatial support.
>
> > "The spatial support of the Canonical ICA used in the current study is shown in Figure 16. Since each spatial component extends across the whole brain, each component is displayed in a glass brain representation, from three different perspectives."
>
> - An analysis of whole brain forecasts:
>
> >   "In Figure 17 a comparison of the error in future brain activity forecast between the original GOKU-net and GOKU-UI is presented. Color code represents the RMSE between the actual brain activity and the forecasts beyond the reconstruction limit generated by these models. In this full brain visual representation it is also noticeable that GOKU-UI generates better forecasts than its original GOKU-net counterpart. However, the deterioration of the forecast across the brain is not homogeneous in either case. Actually, both in the forecasts and reconstructions (see Figure 4}) we notice that an accentuated error tends to be present in the region of precuneus and posterior cingulate cortex. This area is a critical hub of the default mode network and is particularly associated with reflective self-conscious thought and internally focused cognitive activities (Whitfield-Gabrieli et al., 2011). In this sense, the neural dynamics of the precuneus and posterior cingulate cortex region are less trivially entrained by the environment because they play a central role in the trafficking of information between all the other regions of the brain (Donnelly-Kehoe et al., 2019). As a consequence, they are key parts of the "dynamical core" of the brain, making their activity particularly difficult to predict, especially during resting state (Deco et al., 2017)."

---

> ### Author Response · Authors · 2023-12-13
> **Response (2 of N)**
>
> 3. Accessibility for a Broader Audience: Addressing your concern about the paper not being amenable to a large audience, we have added an explanatory section at the beginning of Section 2.1 (page 3):
>
>
> > “GOKU-nets can be considered a specific instance within the broader category of models known as Latent Differential Equation models (Latent DEs). These generative models have a structure similar to Variational Autoencoders (VAEs), but with a key distinction: they encode time series data into a latent space governed by differential equations. The objective, akin to any autoencoder model, is to ensure that the output closely reconstructs the input by passing through this latent space. In Latent DEs, the input typically comprises complex, high-dimensional time series data, such as sequential brain images, financial records, or biosignals. After encoding the data into its latent space, beyond simply generating an output that corresponds to the time span of the input, these models can integrate the differential equation further in time, allowing them to forecast the system's future evolution. This capability is not only crucial for validating a model, as is standard in scientific modeling, but also has diverse practical applications. These applications range from finance and weather forecasting to neuromodulation in the context of brain data, where accurate short-term forecasts can help compensate for delays in closed-loop feedback systems (Parastarfeizabadi & Kouzani, 2017).”
>
> &nbsp;
>
>
> In closing, we appreciate the opportunity to enhance our manuscript based on your valuable feedback. We hope that these revisions and additions address your requested changes effectively and believe they have improved the overall quality of our work.
>
> Sincerely,
>
> The Authors.

---

### Review · Reviewer_jRpQ · 2023-11-21

**Summary Of Contributions:**

This paper proposed a new SciML generative model GOKU-UI, building on the basis of the Latent ODEs, for modeling scientific time-series data.  Compared with previous methods,  GOKU-UI incorporates attention mechanisms and multiple shooting training strategy in the latent space, leading to improved empirical performance in reconstruction and forecast tasks.

**Audience:**

Yes

**Claims And Evidence:**

Yes

**Requested Changes:**

Please refer to above detailed comments in weakness.

**Strengths And Weaknesses:**

The paper has several strengths. It comprehensively explored and evaluated the GOKU-UI model for reconstructing and forecasting time series data. The writing of the paper is very clear, and the detailed information such as the model architecture, training procedures and hyperparameter settings are clearly given. And the experiment includes both synthetic and empirical fMRI datasets.

Weaknesses:

1. Compared with GOKU-net, the proposed enhancements are quite incremental. The authors updated the GOKU network backbone by attention mechanism and incorporated multiple shooting training strategies for GOKU-net training. The attention mechanism and multiple shooting training strategy are all well-established in Neural ODE areas. Although some improvements may come from better backbone and training strategy,  the modifications are minor and no valuable insights are provided.

2. The authors claimed that a penalty term has been added into the training objective to regularize the continuity of trajectories across different windows for multiple shooting. The penalty term needs to be better motivated and clarified, as this is the core difference between the proposed training strategy and the original multiple shooting training strategy. From my current understanding,  the continuity may be distributed during the training,  with these soft loss constraints.

3. Experiment results lack enough comparison with GOKU-net.  In GOKU-net [Linial et al. (2021)],  they have conducted comprehensive evaluations on realistic data, including single pendulum reconstruction from pixels, double pendulum reconstruction from pixels and cardiovascular system parameters identification. In this paper, the method was only evaluated on one fMRI forecast task. We cannot draw a confident conclusion whether this method is actually effective.

4. The paper compares the proposed method with GOKU and naive latent Neural ODE. However, it does not compare the performance of the framework with other NODE models specifically designed for complex time-series data. For example, ANODE, which augmented the ODE with a vector of zeros, is a very effective baseline in tackling complex time series data. Comparison with those methods would provide a better understanding of the strengths and weaknesses of the proposed framework in relation to similar approaches.

5. The proposed enhancements can not demonstrate consistent improvements in forecasting with different time intervals. The prediction error becomes more sensitive to the prediction length compared with latent ODE and GOKU baselines. Also latent ODE and naive LSTM can outperform the proposed method by a large margin for 15 - 20 seconds prediction of brain activity.

---

> ### Author Response · Authors · 2023-12-13
> **Response (1 of N)**
>
> Thank you for your dedication and time spent reviewing our work.
>
> 1. We recognize your concerns about the incremental nature of the improvements to the original GOKU-nets model and the use of the multiple shooting method, which has started to emerge in Neural ODEs, continuing the long tradition of systems identification and control. However, what we present in this work is an extension of this method to latent-variable models and particularly for GOKU-nets, where the multiple shooting method had never been employed before. Moreover, we consider the improvements to be not minor, but rather significant and with large effect sizes. These were calculated by comparing the original GOKU-nets against GOKU-UI:
>
> &nbsp;
>
>
> **Stuart-Landau Simulated - Reconstruction Task**
>
> Average Improvement: 3x
>
> | Training Samples | Cohen’s d Effect Size         |
> |------------------|---------------------|
> | 75               | 16.79 [11.51, 22.07]|
> | 150              | 7.8 [5.23, 10.37]   |
> | 300              | 7.82 [5.24, 10.4]   |
> | 600              | 4.75 [3.04, 6.47]   |
> | 1200             | 5.52 [3.6, 7.45]    |
> | 2400             | 3.29 [1.94, 4.63]   |
> | 4800             | 3.86 [2.38, 5.35]   |
>
> &nbsp;
>
> **Stuart-Landau Simulated - Forecast Task**
>
> Average Improvement: 2x
>
> | Training Samples | Cohen’s d Effect Size         |
> |------------------|---------------------|
> | 75               | 6.38 [4.22, 8.54]   |
> | 150              | 8.75 [5.9, 11.6]    |
> | 300              | 6.16 [4.06, 8.26]   |
> | 600              | 4.08 [2.54, 5.62]   |
> | 1200             | 5.26 [3.41, 7.12]   |
> | 2400             | 2.91 [1.65, 4.16]   |
> | 4800             | 3.27 [1.93, 4.61]   |
>
> &nbsp;
>
> **Empirical fMRI - Reconstruction Task**
>
> Average Improvement: 5x
>
> | Training Samples | Cohen’s d Effect Size           |
> |------------------|-----------------------|
> | 246              | 59.18 [37.98, 80.38]  |
>
> &nbsp;
>
> **Empirical fMRI - Forecast Task (Time in seconds)**
>
> | Time | Cohen’s d Effect Size           |
> |------|-----------------------|
> | 3    | 32.01 [20.51, 43.5]   |
> | 6    | 19.91 [12.71, 27.11]  |
> | 9    | 10.4 [6.54, 14.26]    |
> | 12   | 4.57 [2.64, 6.49]     |
> | 15   | 1.44 [0.3, 2.58]      |
> | 18   | 0.41 [-0.61, 1.44]    |
> | 21   | 0.31 [-0.72, 1.33]    |
> | 24   | 0.48 [-0.55, 1.51]    |
> | 27   | 0.72 [-0.32, 1.77]    |
> | 30   | 0.97 [-0.1, 2.04]     |
>
> Therefore, we could conclude that the improvements due to the incorporation of attention and multiple shooting into GOKU-nets are large (d > 0.8, according to the criterion suggested by Cohen [1]) in all cases except for fMRI future predictions beyond 15 seconds. We have added the effect sizes in the Supplementary Information section D and included the following text in Discussion section:
>
> > The GOKU-UI, while an extension of GOKU-nets rather than a completely new model, significantly enhances performance by integrating attention and multiple shooting. This has resulted in large effect sizes across various applications, as detailed in Supplementary Tables 1 and 1. The notable exception is in fMRI future predictions beyond 15 seconds, where these enhancements are less effective.
>
>
> 2. Regarding the penalty method, it is important to clarify that this is not the core difference between the proposed training strategy and the original multiple shooting training strategy. Instead, the penalty method is a standard approach to enforce continuity between multiple temporal windows in multiple shooting, aiming to make the set of trajectories equivalent to a trajectory obtained via single shooting. This approach has already been used in the context of Neural ODEs, as indicated in references [2, 3, 4]. The novelty in our method lies in extending multiple shooting to a latent-space model. Regarding the distribution of continuity during training, if you refer to modifying the penalty parameter throughout training, it is possible to increase it incrementally. However, in our experiments, we kept it fixed during all trainings.
>
>    We have added references after explaining the penalty method in Section 2.2.2 (page 5) to indicate that it is not a unique innovation of ours but is based on previous literature:
>
> > Specifically, we employ regularization in the cost function when training the model, quadratically penalizing the discrepancy in the latent space of the overlapping points, that is, between the initial condition of each window and the end point of its preceding segment (Vantilborgh et al., 2022; Turan & Jäschke, 2021).
>
>  We also appreciate your pointing out the lack of motivation for enforcing continuity between the windows of multiple shooting. In the same section, we added:
>
> > As mentioned before, we do not have access to the true initial conditions for grounding the latent trajectories. However, we can strive to achieve continuity across different windows, which is crucial in multiple shooting methods to obtain solutions equivalent to those achieved with single shooting ones.

---

> ### Author Response · Authors · 2023-12-13
> **Response (2 of N)**
>
> 3. We regret any confusion caused by our presentation. In our paper, we did not limit our evaluation to only one fMRI forecast task. Instead, we conducted comprehensive evaluations on both synthetic and empirical data. Specifically, our experiments included tasks on simulated data based on Stuart-Landau oscillators for both reconstruction and forecast tasks, as well as on empirical data derived from fMRI, again covering both reconstruction and forecast tasks.
>
>    This dual approach, involving both synthetic and empirical datasets, was designed to thoroughly assess the performance and applicability of our model across different scenarios. The synthetic data, based on Stuart-Landau oscillators, provided a controlled environment to test the model's capabilities in handling complex dynamical systems. The empirical fMRI data, on the other hand, offered a real-world application scenario, allowing us to evaluate the model's practical utility in reconstructing and forecasting time-series data derived from human brain activity.
>
> &nbsp;
>
> 4. Thank you for your valuable suggestion regarding the inclusion of the  Augmented Neural ODE model as an additional baseline for comparison. In response to your feedback, we have updated our manuscript to incorporate ANODE into our set of baseline models.
>
> &nbsp;
>
> 5. Thank you for highlighting the variation in the performance of GOKU-UI across different forecasting horizons. We acknowledge that the fluctuating efficacy in longer-term predictions is an inherent characteristic of the model, which should be considered in its application. Indeed, beyond the 15-second forecasting window, GOKU-UI's predictions may not be as robust as some of the baseline models, though it still performs marginally better than the basic GOKU-nets with single shooting. However, it's important to note that at these longer forecast intervals, all models, including ours, tend to exhibit decreased performance, which aligns with expectations given the complexity of systems like the human brain.
> Moreover, in many practical applications, such as neuromodulation [5], where accurate short-term forecasts are crucial for compensating delays in closed-loop feedback systems, long-term predictions are less critical as short-term forecasts are recalculated repeatedly. This context underscores the relevance and utility of our model in scenarios where short-term forecasting is paramount.
>
>    In recognition of this limitation, we have added the following text to the end of Section 3.2 (page 9) to explicitly acknowledge this aspect of our model:
>
> > Beyond this point, GOKU-UI was outperformed by other baseline models, though it continued to exhibit marginally better performance compared to the basic GOKU-nets with single shooting. Nonetheless, all models demonstrated limited efficacy in long-range forecasts, a foreseeable outcome considering the complexity of the system under study. Given their exclusive training on reconstruction tasks, these models present opportunities for further refinement and enhancement.
>
> This addition aims to provide a comprehensive understanding of our model's capabilities and limitations, especially in the context of forecasting over different time intervals. We believe that this acknowledgment not only enhances the transparency of our findings but also opens avenues for future improvements and applications of the model.
>
> &nbsp;
>
>
> We are grateful for your detailed and helpful feedback, which has significantly contributed to refining and improving our manuscript.
>
> &nbsp;
>
> References:
>
> [1] Cohen J. (1988). Statistical Power Analysis for the Behavioral Sciences. New York, NY.
>
> [2] Vantilborgh, V., Lefebvre, T., and Crevecoeur, G. Efficient ODE Substructure Identification of the Acrobot under Partial Observability using Neural Networks and Direct Multiple Shooting. In 2022 IEEE/ASME International Conference on Advanced Intelligent Mechatronics (AIM), pp. 1263–1268. IEEE, 2022.
>
> [3] Turan, E. M. and Jäschke, J. Multiple shooting for training neural differential equations on time series. IEEE Control Systems Letters, 6:1897–1902, 2021.
>
> [4] https://docs.sciml.ai/DiffEqFlux/stable/examples/multiple_shooting/
>
> [5] Parastarfeizabadi, M. and Kouzani, A. Z. Advances in closed-loop deep brain stimulation devices. Journal of neuroengineering and rehabilitation, 14(1):1–20, 2017.

---

### Decision · Action_Editor_NGzz · 2024-01-30

**Recommendation:** Accept as is

**Comment:**

This paper utilizes attention mechanisms and multiple shooting training strategy in the latent space to build on GOKU-net. The resulting GOKU-UI is benchmarked on reconstruction and forecasting tasks. The authors were able to resolve the concerns that reviewers had with the initial submission, to argue that GOKU-UI is a substantial improvement over GOKU-nets and the evaluation benchmarks are performed without label-leakage.

**Audience:**

Researchers interested in neural ODE's and SciML generative models would find this paper relevant.

**Claims And Evidence:**

This paper introduces GOKU-UI, a SciML generative model that claims to significantly outperform GOKU-net. The authors have provided a substantial number of experiments by the end of the rebuttal phase that supports this claim. The reviewers all agreed by the end of the rebuttal period.